# A fine-grained evaluation framework for urban land cover change based on feature monitoring with remotely sensed imagery

Qiang Liu[1], Jiachen Guo[1], Chuanxing Zheng[2]*, Feng Ling[3], Zhixiang Da[4], Wenlong Song[5], Fengjiao Zhao[1], Jijian Lian[1]

1 School of Ocean Energy, Tianjin University of Technology, Tianjin, China, 2 Tianjin Huashui Engineering Consulting Co., Ltd, Tianjin, China, 3 Soil and Water Conservation Monitor Center, HaiHe Basin Remote Sensing Monitor ET Center, HWCC, Tianjin, China, 4 Soil and Water Conservation Workstation, Tianjin Water Authority, Tianjin, China, 5 State Key Laboratory of Simulation and Regulation of Water Cycle in River Basin, China Institute of Water Resources and Hydropower Research, Beijing, China

* 13021326281@163.com

## Abstract

Against the backdrop of accelerating global climate change and urbanization, urban land cover change has emerged as a critical indicator for understanding the dynamic evolution of cities and the transformation of urban ecosystems. This study proposes a data-driven framework for fine-scale urban land cover change assessment based on the UASFNet model, enabling high-precision evaluation of urban land cover dynamics. The approach first performs preprocessing and co-registration of bi-temporal remote sensing images from the study area, and applies the trained UASFNet model to identify urban land cover types and extract land cover information for each temporal phase. The Analytic Hierarchy Process (AHP) is then employed to determine the weights of various indicator factors. By integrating building disturbance, greenbelt disturbance, and road disturbance indices, the framework quantitatively evaluates the intensity of land cover change at both pixel and regional scales. Experimental results across three benchmark datasets, consisting of high-resolution sub-meter RGB urban remote sensing imagery, demonstrate that UASFNet achieves superior segmentation accuracy, with mean Intersection over Union (mIoU) values of 91.52%, 93.31%, and 88.90%, substantially outperforming several state-of-the-art baseline models. Spatial analysis of the Langfang urban area (2017–2023) reveals a marked increase in impervious surface coverage (+16.86%) and a sharp decline in greenbelt (−40%), with the urban landscape exhibiting a multi-core, belt-like expansion pattern oriented toward newly developed districts. The proposed framework not only enhances the interpretability and generalization of remote sensing models in complex urban environments but also provides a scalable analytical tool to support urban spatial planning, ecological conservation, and sustainable city governance.

**Data availability statement:** Due to data use agreements and institutional restrictions, the datasets generated and/or analyzed during the current study are not publicly available. Data access requests may be directed to the Data Management Office of Tianjin Huashui Engineering Consulting Co., Ltd. (contact email: tjhuashui_data@163.com), which is independent of the authors and responsible for coordinating data access and ensuring long-term data stewardship on behalf of the institution. Access will be considered upon reasonable request and subject to institutional approval.

**Funding:** The research was supported by Tianjin major science and technology projects (24ZXKJGX00070) awarded to Q.L., National Key R&D Program of China (2022YFB4200700) awarded to Q.L., and Research and Application of Key Technologies for Intelligent Mechanical Manufacturing Production Lines (23ZGZNGX00020) awarded to Q.L.

**Competing interests:** the commercial affiliation of some authors with Tianjin Huashui Engineering Consulting Co., Ltd. The revised statements clarify that the funder provided support in the form of salaries only and had no role in the study design, data collection and analysis, decision to publish, or preparation of the manuscript. We also confirm that this affiliation does not alter our adherence to all PLOS ONE policies on sharing data and materials.

## 1. Introduction

With the continued intensification of global urbanization and climate change, the increasing frequency of extreme weather events has become one of the most prominent environmental challenges [1–4]. Observational evidence indicates that global mean temperature has risen by approximately 1.1 °C over the past century [5], accompanied by a significant increase in both the intensity and frequency of extreme precipitation events [6,7]. Against this backdrop, impervious surfaces within urban areas have expanded rapidly, profoundly altering the physical structure of regional underlying surfaces [8–12]. According to the United Nations World Urbanization Prospects (2022), the global proportion of urban population has increased from 30% in 1950 to 57% in 2020 and is projected to exceed 68% by 2050 [13]. The urbanization process in China has been particularly remarkable, with the urbanization rate soaring from 17.9% in 1978 to around 67% by 2025, while urban construction land area has expanded nearly fivefold [14]. This rapid transformation has led to large-scale replacement of natural surfaces such as greenbelts and wetlands by impervious materials, resulting in a pronounced spatial reorganization of regional land cover structures. Therefore, establishing a scientifically robust and fine-scale urban land cover assessment framework capable of accurately capturing the spatial distribution and temporal evolution of surface cover types has become an urgent need for understanding urban environmental change and advancing spatial governance.

In recent years, urban land cover change has become a central topic in global environmental change and sustainable development research [15–17]. Existing studies have primarily focused on the identification of urban expansion patterns [18,19], the analysis of land-use driving mechanisms [20,21], and the evaluation of landscape evolution dynamics [22–24]. Earlier approaches mainly relied on statistical data or low-resolution remote sensing imagery, using indicators such as land-use transition matrices, landscape metrics, and temporal change rates to characterize urban expansion [25]. However, these methods are limited in both spatial resolution and classification accuracy, making it difficult to capture the fine structural and dynamic features of complex urban surfaces [26]. With the rapid advancement of high-resolution remote sensing data and deep learning technologies, research efforts have increasingly shifted toward fine-grained classification and dynamic monitoring of urban surfaces using multi-temporal, multi-scale, and multi-source imagery [27–30]. In particular, the introduction of deep learning methods has greatly enhanced the automation and accuracy of land-cover feature recognition [28,31]. Although models such as UNet [32], DeepLab [33], and SegFormer [34] have achieved notable progress in remote sensing image segmentation, most existing studies remain focused on natural surfaces or general suburban areas [35–39]. Fine-scale identification and change detection within typical urban cores—characterized by dense building clusters, intricate road networks, and interwoven greenbelts—are still relatively underexplored.

Urban land cover change not only reflects the rate and direction of spatial expansion but is also directly linked to the evolution of ecological security patterns and

the vulnerability of urban systems [11,40,41]. In existing studies, mainstream assessment approaches typically rely on indicator-based frameworks and weighting analyses [42]—employing techniques such as the AHP [43], Entropy Weight Method [44], and Fuzzy Comprehensive Evaluation [45]—to quantify the magnitude of land-use change and its ecological implications. These methods have been widely applied in regional-scale analyses of urban vulnerability, sustainability, and landscape pattern dynamics [46]. However, conventional approaches that depend on statistical data or low spatiotemporal resolution imagery are insufficient to capture subtle surface disturbances and spatial heterogeneity associated with urbanization processes. By leveraging pixel-level land cover change results derived from deep learning models as the primary analytical driver, a fine-scale urban surface change assessment framework can be established to accurately identify and quantify urban land cover dynamics. However, existing studies often treat fine-grained land cover mapping and change assessment as relatively independent tasks. Deep learning–based segmentation models are mainly evaluated in terms of pixel-level accuracy, while indicator-based change assessment frameworks commonly rely on coarse-resolution or aggregated inputs. As a result, the linkage between boundary-aware urban land cover recognition and disturbance-level interpretation remains weak, particularly in dense urban cores. This gap highlights the need for an integrated framework that explicitly connects fine-scale semantic segmentation with weighted, interpretable urban land cover change assessment.

This study aims to propose a data-driven framework for fine-scale urban land cover change assessment based on UASFNet, designed to support urban spatial governance and ecological optimization decisions while revealing the dynamic evolution of surface systems during urbanization. The potential academic contributions of this study are as follows: (i) An Urban Adaptive Shared-feature Attention Network (UASFNet) is developed for fine-grained urban land cover mapping, and its effectiveness in enhancing boundary delineation and inter-class separability is systematically evaluated against representative semantic segmentation baselines in complex urban scenes;(ii) A weighted disturbance integration scheme based on the Analytic Hierarchy Process (AHP) is introduced to fuse building, road, and greenbelt change indicators, enabling interpretable and spatially coherent urban land cover change assessment;(iii) A multi-scale evaluation framework combining pixel-level classification and region-level aggregation is proposed to improve the characterization and interpretability of urban land cover change patterns beyond pixel-only analyses. The structure of this paper is organized as follows: The Materials and Methods section describes the materials and methods, including the proposed methodological framework and data preparation. The Results section presents the results of urban land cover classification and change analysis based on the UASFNet model. The Discussion section discusses the performance, applicability, and implications of the proposed approach. The Conclusion section concludes the study.

## 2. Materials and methods

### 2.1. Fine-scale assessment framework for urban land cover change

This study proposes a data-driven framework for fine-scale urban land cover change assessment based on the UASFNet architecture (Fig 1). The framework consists of four main stages. First, bi-temporal high-resolution remote sensing images of the study area are preprocessed and co-registered to ensure spatial consistency between different observation periods. These images serve as the input data for pixel-level land cover classification. Second, the trained UASFNet model is applied to the preprocessed imagery to identify urban land cover types and generate fine-grained land cover maps for each temporal phase. The resulting classification outputs provide the basis for subsequent change analysis. Third, building disturbance, greenbelt disturbance, and road disturbance indicators are derived from the classification results. The Analytic Hierarchy Process (AHP) is employed to determine the relative weights of these indicators, enabling the integration of heterogeneous change information. Finally, the weighted disturbance indicators are aggregated to evaluate the comprehensive intensity and spatial distribution of urban land cover change across the study area.

### 2.2. Data preparation

#### 2.2.1. Study area. In the experiments, to evaluate the segmentation performance and computational efficiency of the proposed algorithmic model, a comparative study was conducted using three high-resolution RGB image datasets

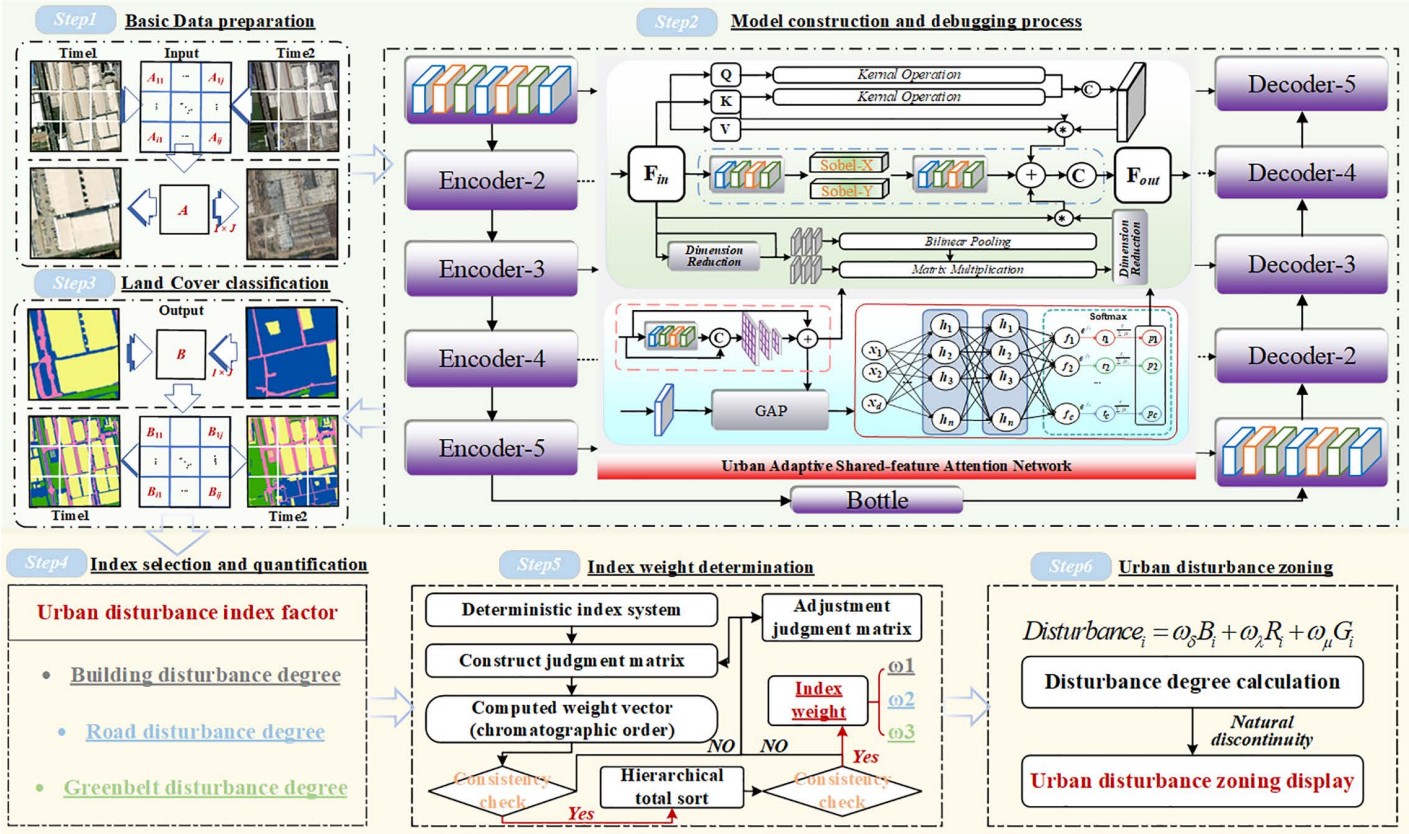

**Fig 1. Overall framework of the proposed fine-scale urban land cover change assessment method. All elements were created by the authors.**

obtained from different geographic regions and satellite sources: (a) the Langfang dataset, (b) the Potsdam dataset, and (c) the Guiyang dataset. Fig 2 illustrates the geographical locations of the study areas and the corresponding images for the three datasets.

**2.2.2. Database construction.** Fig 3 presents sample images and corresponding semantic label maps from the Potsdam datasets. The detailed parameters of each dataset are as follows:

(a) **Langfang Dataset**: This dataset was constructed using RGB three-band remote sensing images of the Langfang Economic Development Zone with a spatial resolution of 0.6 m. Four land-use classes were annotated using GIS software, and the dataset was partitioned into training, validation, and test sets using a spatial block segmentation method at a ratio of 70%/ 10%/ 20%, respectively.

(b) **Potsdam Dataset**: This dataset is derived from ultra–high-resolution TOP imagery with a ground sampling distance (GSD) of 5 cm. The Potsdam region is known for its complex building layouts and dense urban structures. The dataset covers an area of 3.42 km² and includes pixel-level annotations for six semantic categories. It has become a standard benchmark for semantic segmentation research. In this study, an improved four-class semantic labeling scheme— including buildings, greenbelts, roads, and others—was adopted to better suit urban analysis tasks, resulting in 2299 training, 605 validation, and 1694 test image patches of size 1024 × 1024.

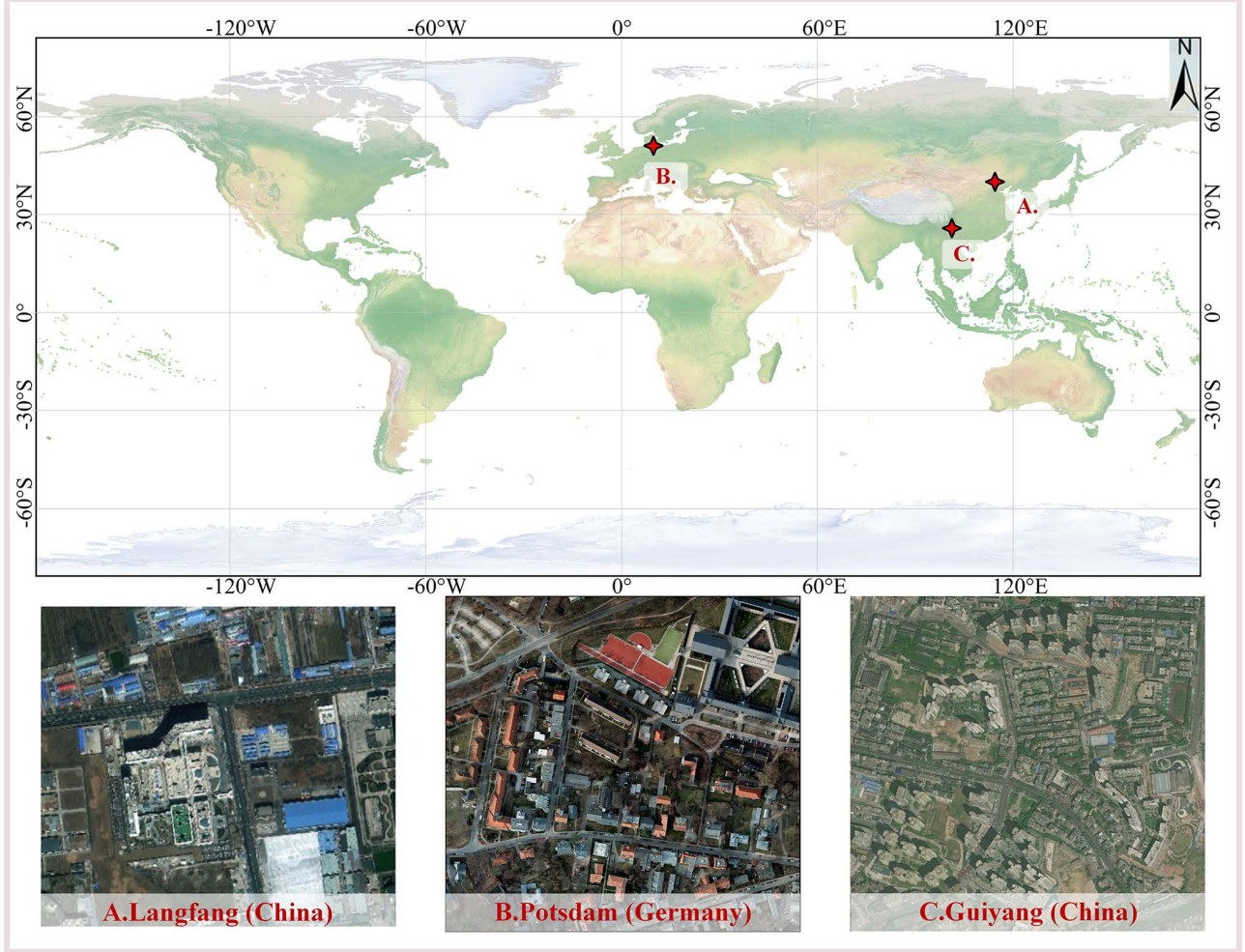

**Fig 2. Location map of the study area.** (A) the Langfang dataset (China). (B) the Potsdam dataset (Germany). (C) the Guiyang dataset (China). The global location map was generated using the Natural Earth dataset, which is in the public domain. Representative remote sensing imagery was obtained from the Copernicus Data Space Ecosystem (Copernicus Sentinel data), the ISPRS Potsdam dataset and processed by the authors. All map elements and annotations were created by the authors.

(c) **Guiyang Dataset**: This dataset covers the urban core area of Guiyang City, China, representing a typical mountainous urban environment. It was constructed following the same data preparation, annotation protocol, and data partition strategy as the Langfang dataset, ensuring consistency and comparability across datasets.

For all datasets, standard preprocessing procedures were applied prior to model training and inference to ensure consistency across multi-source and multi-temporal imagery. These procedures include geometric co-registration between different acquisition periods, spatial resampling to a unified spatial resolution, and normalization of RGB values. No aggressive radiometric correction or handcrafted feature extraction was introduced, in order to preserve original spatial patterns. The same preprocessing pipeline was applied to all datasets and baseline models to ensure a fair and unbiased comparison [47,48]. All remote sensing imagery used in this study was obtained from publicly accessible datasets under open data policies, including Copernicus Sentinel data and the ISPRS Potsdam dataset. All maps and visualizations were generated by the authors. No third-party online basemap services were used.

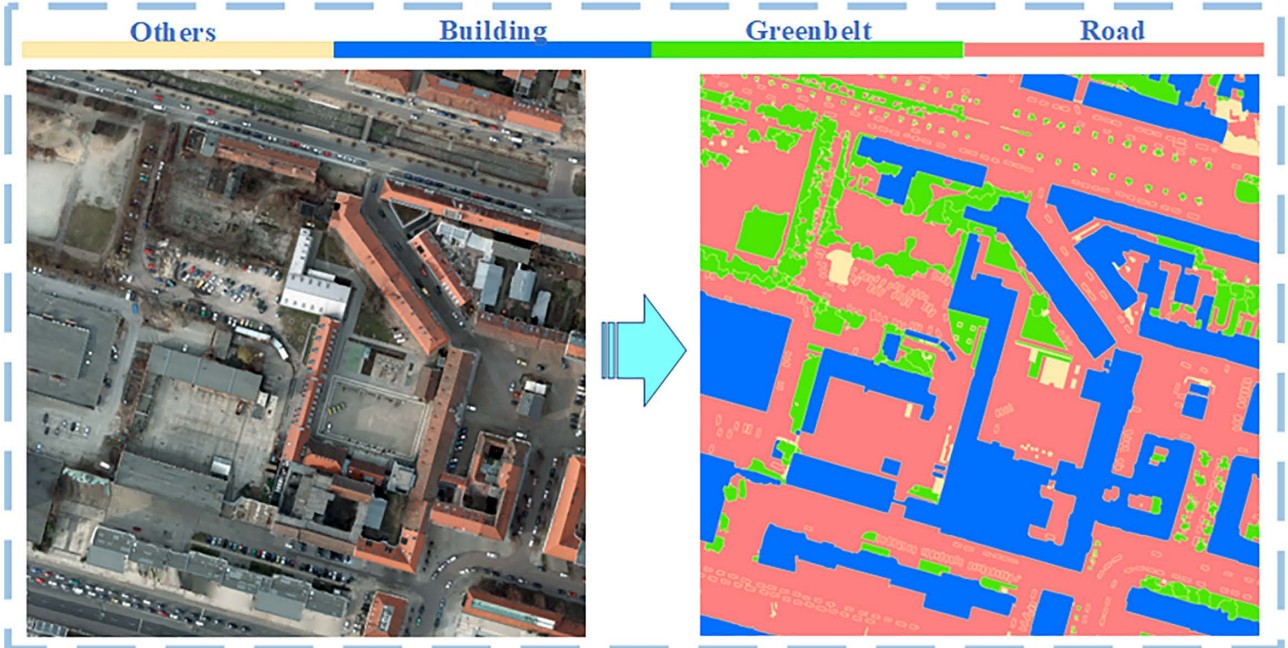

**Fig 3. Example image patches and corresponding ground truth labels from the Potsdam dataset. The figure was generated by the authors for visualization purposes.**

### 2.3. Structure of the urban adaptive shared-feature attention network (UASFNet)

In urban land cover change detection, different surface objects exhibit pronounced spatial heterogeneity and semantic coupling relationships. Urban features such as buildings, roads, and greenbelts differ substantially in spatial scale and morphological characteristics, and their spectral representations in imagery often display strong local discontinuities and abrupt boundary transitions. To address these challenges, this study proposes an Urban Adaptive Shared Feature Attention (UASF) module that achieves adaptive fusion of semantic consistency, spatial continuity, and structural stability through a unified mapping mechanism (See S1 Table).

Given an input feature $X \in R^{B \times C \times H \times W}$, the UASF module can be formulated within the joint domain $Z = \aleph \otimes \vartheta$ of the feature space $\aleph$ and the geometric space $\vartheta$ as follows:

$$F_{UASF}(X) = \Phi_{proj}(A(X) \odot K_{UASF}(X)) + X \tag{1}$$

Here, $K_{UASF}(X)$ denotes the semantic–structural consistency kernel mapping, $A(X)$ represents the global adaptive gating operator, and $\Phi_{proj}(\cdot)$ refers to the channel reconstruction and projection mapping. This formulation defines the overall mapping relationship of the module in the form of operator composition, enabling UASF to achieve unified modeling of semantic aggregation and dynamic modulation within the joint feature–structure space. Unlike conventional residual structures, the joint mapping mechanism of UASF not only performs nonlinear feature reconstruction but also establishes a synergistic linkage between semantic and geometric representations at the operator level.

First, the core of the UASF module is the semantic–structural consistency kernel function $K_{UASF}(X)$, which simultaneously computes correlations within both the feature domain and the gradient domain to capture long-range semantic dependencies and local boundary characteristics. Its formal representation is given as:

$$K_{UASF}(X) = \text{Softmax}\left[\phi(X)\phi(X)^{\mathsf{T}} \oplus \varphi(\nabla X)\right]\Theta(X) \tag{2}$$

Here, $\phi(X)$ denotes the shared semantic embedding obtained through depthwise separable convolution, which is used to extract spatial contextual features; $\varphi(\nabla X)$ represents the structure modulation function based on the Sobel gradient operator, designed to characterize object boundary information; $\Theta(X)$ is the linear reconstruction operator responsible for generating the fused feature representation.

After completing the joint modeling of semantic and structural features, the UASF module adaptively balances the contribution weights of different feature domains through a global gating mechanism. The gating operator $A(X)$ is defined as:

$$A(X) = \Gamma_{gate}(GAP(X)) = \text{Softmax}\left[W_2\sigma(W_1 GAP(X))\right] \tag{3}$$

Here, $GAP(X)$ denotes the global average pooling operation, $\Gamma_{gate}(\cdot)$ represents the dynamic gating function composed of two fully connected layers, $\sigma(\cdot)$ is the nonlinear activation function, and $W_1$ and $W_2$ are the learnable weight matrices. Through this mechanism, the model dynamically adjusts the weighting between semantic and structural features during the fusion process according to the global feature distribution, thereby achieving adaptive responses under varying levels of spatial complexity.

The semantic–structural consistency kernel encourages the alignment of semantic representations with geometric cues, which is beneficial for preserving object boundaries and reducing confusion between visually similar land cover classes. The gradient-domain formulation highlights local structural variations, such as edges and shape transitions, that are particularly relevant for fine-scale urban patterns. The global gating operator adaptively modulates the relative contributions of semantic and structural information across spatial locations, thereby promoting spatial coherence while limiting redundant feature propagation.

At the architectural level, the UASF module is embedded into the bottleneck and decoder stages of the U-Net backbone, forming the core framework of UASFNet. By aggregating global semantic features during the encoding phase and restoring spatial boundary details during the decoding phase, UASFNet establishes a robust adaptive coupling between global and local representations. This design not only effectively mitigates issues of object overlap and boundary discontinuity commonly observed in high-resolution imagery but also provides, from a theoretical perspective, an interpretable end-to-end modeling mechanism of semantic–geometric consistency. In summary, under a unified mechanism, UASFNet achieves coordinated optimization of global modeling, structural constraint, and adaptive fusion, significantly enhancing boundary delineation accuracy and semantic representation in high-resolution urban remote sensing scenes, thereby providing a stable feature foundation for subsequent change detection and spatiotemporal disturbance analysis.

## 2.4. Evaluation metrics

This study conducts a comprehensive evaluation of the land classification model's performance using multiple metrics to ensure the objectivity and reliability of the results. The calculation method is shown in Table 1 [49]. The mean Intersection over Union (mIoU) was used as the primary metric to quantify overall pixel-level classification accuracy across land-cover classes. Precision, recall, and F1-score were employed to further characterize class-wise discrimination performance, particularly for imbalanced or structurally complex categories. In addition, Kappa coefficient was included to evaluate the agreement between predicted and reference maps beyond chance, providing a robust measure of classification reliability. Together, these metrics capture not only pixel-wise accuracy but also classification consistency and robustness, which are essential for ensuring that the resulting land-cover maps can serve as reliable inputs for subsequent urban change and disturbance analysis.

Here, $TP$, $TN$, $FP$, and $FN$ represent true positives, true negatives, false positives, and false negatives, respectively. C is the total number of categories, and F1i is the F1 Score for the i-th category. Precision represents the proportion of

**Table 1. Evaluation index.**

| Effect | Evaluation index and formula |
|---|---|
| Overall accuracy evaluation | $Accuracy = \frac{TP+TN}{TP+TN+FP+FN}$ (4) |
| Classification ability assessment | $Precision = \frac{TP}{TP+FP}, Recall = \frac{TP}{TP+FN}$ (5) |
| | $MF1 = \frac{1}{C}\sum_{i=1}^{C} \frac{2 \times Precision_i \times Recall_i}{Precision_i + Recall_i}$ (6) |
| Prediction region matching degree evaluation | $IoU = \frac{TP}{TP+FP+FN}, MIoU = \frac{1}{C}\sum_{i=1}^{C} IoU_i$ (7) |
| Consistency evaluation | $K = \frac{P_0-P_e}{1-P_e}$ (8) |

actual positive samples among those predicted as positive. Recall represents the proportion of actual positive samples correctly identified as positive.

It should be noted that, as in most large-area urban land cover mapping studies, reference labels are subject to unavoidable uncertainties, particularly in transitional zones, complex building boundaries, narrow road segments, and shadow-affected areas. Since all models were trained and evaluated using the same reference data and partitioning strategy, such uncertainties affect all methods consistently and do not compromise the validity of the relative performance comparison.

## 2.5. Methods for urban land cover change assessment

This study proposes an urban land cover (LC) change assessment framework that integrates the advantages of image-based recognition and indicator system analysis. The specific steps are as follows: (1) The urban land cover recognition module is used to construct hierarchical layers for building disturbance, road disturbance, and greenbelt disturbance, upon which a comprehensive LC change evaluation indicator system is established. (2) AHP is then applied to calculate the weights of each indicator, enabling quantitative evaluation and hierarchical representation of urban land use value changes. Direct aggregation of these disturbance indicators would implicitly assume equal importance across different land-cover components, which may mask dominant drivers of urban land cover change and reduce interpretability in dense urban environments.

### 2.5.1. Selection and quantification of indicator factors. The selection and quantification of index factors are mainly achieved through the following steps:

(1) Framework of index system

The structure of the indicator system in this study is mainly divided into a target layer and an indicator layer. The target layer represents the urban LC change assessment system. The indicator layer consists of the urban building disturbance layer, urban road disturbance layer, and urban greenbelt disturbance layer.

(2) Large-scale disturbance layer preparation

First, the satellite images are segmented using a sliding window method, dividing high-resolution remote sensing images from different time periods into multiple pixel-based image patches, each representing an independent spatiotemporal unit. Next, the pre-trained urban land cover classification model is applied to perform pixel-level multi-class classification on each image patch, and the classification results are stored in raster file format. Finally, the image patches are reassembled according to the original mapping scheme to reconstruct a complete regional land cover classification raster map, as well as individual raster layers for specific land-use types, for subsequent analysis and evaluation.

**(3) Quantization of index factors**

This study uses raster cells as the basic unit to quantify the disturbance areas of buildings, roads, and greenbelts. First, a raster layer with a 10×10 m resolution is generated within the study area, and the disturbance areas of different land types are assigned to the corresponding raster cells. The disturbance area is derived by calculating the difference in land type coverage within the same raster cell at different time points, thereby quantifying the LULC changes in urban land use.

### 2.5.2. Urban LULC changes calculation method.

(1) Index weight calculation method

In this study, weighting is introduced to explicitly reflect the differentiated contributions of building, road, and greenbelt disturbances to the overall urban land cover change assessment. The Analytic Hierarchy Process (AHP) is a mathematical method used for multi-criteria decision analysis, suitable for decision problems with multiple objectives or complex characteristics [50]. This method analyzes complex problems in depth and makes mathematical decisions based on limited quantitative information. In this study, AHP is applied for the quantitative analysis of urban LULC changes factors. The judgment matrix was constructed based on aggregated expert pairwise comparisons, and its logical consistency was verified using the AHP consistency ratio (CR).The specific steps are as follows:

First, each factor is compared pairwise, and a pairwise matrix is established based on the comparison results.

$$M = \begin{bmatrix} 1 & a_{12} & a_{13} & \cdots & a_{1n} \\ a_{21} & 1 & a_{23} & \cdots & a_{2n} \\ a_{31} & a_{32} & 1 & \cdots & \cdots \\ \vdots & \vdots & \vdots & \vdots & \vdots \\ a_{n1} & a_{n2} & \cdots & \cdots & 1 \end{bmatrix}, a_{ij} = \frac{\text{weight of attribute i}}{\text{weight of attribute j}} \tag{9}$$

Where, $a_{ij}$ represents the comparison result of the $i$ th factor with the $j$ th factor, and the value ranges from 1 to 9.

The matrix obtained through calculation can be verified for consistency using the Consistency Ratio (CR), which is calculated using the following formula:

$$CR = \frac{\lambda_{\max} - n}{n - 1} \cdot \frac{1}{RI} \tag{10}$$

Here, $\lambda_{max}$ is the maximum eigenvalue of the matrix, and $n$ is the matrix dimension, and $RI$ is the Random Consistency Index. If $CR$ is less than or equal to 0.1, the judgment matrix has reasonable consistency; if $CR$ is greater than 0.1, the pairwise matrix needs to be adjusted.

In this study, three major factors—buildings, greenbelts, and greenbelt—were considered. After consistency testing, the weight values for each factor were calculated and used for further creation of the urban LULC changes change map.

(2) Disturbance calculation method

This study assesses the overall urban disturbance level by considering the disturbances from urban buildings, roads, and greenbelts. The disturbance calculation formula for each grid cell is as follows:

$$Disturbance_i = \omega_\delta B_i + \omega_\lambda R_i + \omega_\mu G_i \tag{11}$$

In the equation, $B_i$, $R_i$, and $G_i$ represent the urban building disturbance area, urban road disturbance area, and urban greenbelt disturbance area within the $i$-th grid, respectively. $\omega_b$, $\omega_r$, and $\omega_g$ denote the weights of the urban building disturbance index, urban road disturbance index, and urban greenbelt disturbance index, respectively.

### 2.6. Computational environment and data augmentation strategy

In this study, the bi-temporal remote sensing interpretation module was implemented using Python and the PyTorch framework. All model training and evaluation were conducted on a Windows 11 operating system equipped with an NVIDIA® GeForce RTX™ 4090 GPU, ensuring efficient computational resource support. For data augmentation, random horizontal and vertical flipping, as well as rotation within ±15°, were applied to increase the diversity of the training dataset, thereby enhancing the model's ability to generalize to unseen data.

During the training phase, the batch size is set to 4 based on the GPU memory capacity and model complexity, with an initial learning rate of 1e-4. The Adam optimizer is used to optimize the training process by automatically adjusting the gradient mean and variance for each parameter. The total number of training epochs is set to 300 to ensure that the model converges after sufficient learning. In classification, the model is divided into four categories (building, road, greenbelt, and others). A composite loss function combining categorical cross-entropy and Dice loss was adopted, with equal weighting, to balance classification accuracy, improve the recognition of hard samples, and optimize the segmentation of small target regions. This design also helps mitigate edge cases caused by class imbalance and limited training samples for certain land-cover categories, particularly for thin structures such as roads and fragmented greenbelts.

For performance evaluation, UASFNet was compared with several representative deep learning–based semantic segmentation models commonly used in urban land-cover mapping, including ResNet [36], UNetformer [37], CMTFNet [38], CM-Unet [35], and MFNet [39].

## 3. Results

### 3.1. Model performance

This section evaluates the classification performance of the proposed UASFNet in comparison with five models introduced in the Materials and Methods section across different datasets. By reporting both category-level and overall performance, this analysis aims to provide an objective assessment of the relative effectiveness of UASFNet for fine-grained urban land cover classification.

To validate the overall performance of the proposed UASFNet model, comparative experiments were conducted on three representative urban remote sensing datasets—Langfang, Potsdam, and Guiyang—against five representative models. The classification results of each model on the the Potsdam dataset are illustrated in Fig 4. As shown, the spatial recognition performance across different land-cover types varies considerably among models. For the building and road classes, UASFNet accurately preserves the integrity and continuity of object boundaries, producing sharp building edges and coherent road structures. In contrast, models such as CM-Unet and ResNet tend to exhibit blurred boundaries, road discontinuities, or local adhesions in dense urban areas. Regarding greenbelt identification, UNetformer and CMTFNet demonstrate satisfactory overall recognition of greenbelt areas but still produce scattered noise in regions affected by high reflectance or shadow interference. In contrast, UASFNet effectively suppresses such noise through its multi-scale shared-feature attention mechanism, achieving smoother boundary transitions and more coherent greenbelt patches. Overall, across various urban scenes, UASFNet exhibits superior spatial consistency and finer detail restoration, producing segmentation results that more closely approximate real-world land-cover distributions.

Table 2 summarizes the detailed quantitative results of all models across the three datasets. In the Langfang dataset, UASFNet achieved an average mIoU of 91.52%, outperforming the second-best model, CMTFNet, by 1.2%. The IoU values for buildings, greenbelts, and roads reached 96.50%, 95.61%, and 95.12%, respectively—the highest among all models—while the overall Accuracy and Kappa coefficient were 97.29% and 0.9580. On the Potsdam dataset, UASFNet also demonstrated superior performance, achieving an mIoU of 93.31%, an F1-score of 96.37%, and a Kappa coefficient of 0.9783—significantly outperforming MFNet (mIoU 91.96%) and UNetformer (90.55%).On the Guiyang dataset, UASF-Net achieved the best overall performance, with an mIoU of 88.90%, an F1-score of 93.65%, and a Kappa value as high as 0.9493. Among the four land-cover categories, the IoU values for buildings, roads, and greenbelts reached 96.58%,

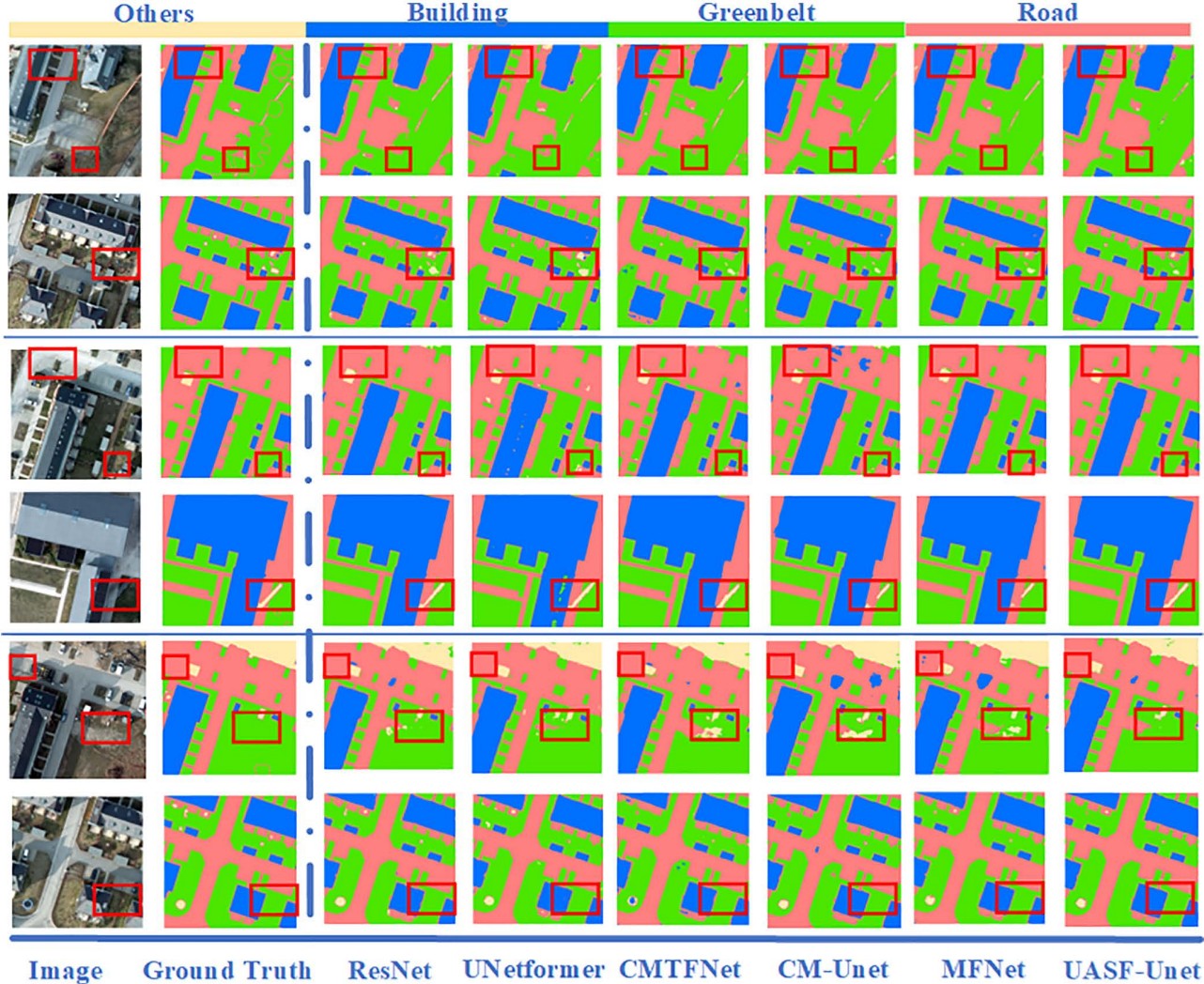

**Fig 4. Visual comparison of urban land cover classification results produced by different models on the the Potsdam dataset. All classification results were generated by the authors.**

96.44%, and 94.62%, respectively. These results demonstrate consistent performance advantages over multiple baseline models and further support the robust generalization capability of UASFNet across different resolutions and domains.

Across the results from the three datasets, UASFNet consistently maintains leading performance on most evaluation metrics, with performance improvements exhibiting strong consistency. Overall, UASFNet exhibited superior performance in boundary preservation, inter-class separability, and overall accuracy compared to benchmark models, confirming that the proposed shared-feature and adaptive attention fusion mechanism effectively enhances multi-scale representation and spatial consistency.

### 3.2. Multi-Temporal urban land cover classification results

This subsection presents the multi-temporal urban land cover classification results derived from UASFNet for different observation periods. By analyzing classification maps from multiple years, this section examines the temporal consistency of the model outputs and highlights major patterns of land cover change associated with urban development.

**Table 2. Quantitative performance comparison of different semantic segmentation models on three urban remote sensing datasets.**

| Dataset | Method | IoU(%) per Category | | | | Metrics | | | |
|---|---|---|---|---|---|---|---|---|---|
| | | Building | Greenbelt | Road | Others | Accuracy | MF1 | MIoU | Kappa |
| Langfang Dataset | ResNet | 95.16 | 91.16 | 87.83 | 63.50 | 94.49 | 91.02 | 84.41 | 0.9153 |
| | UNetformer | 92.78 | 92.49 | 87.56 | 76.54 | 94.70 | 93.11 | 87.34 | 0.9179 |
| | CMTFNet | 96.71 | 92.66 | 92.77 | 79.07 | 96.71 | 94.77 | 90.30 | 0.9277 |
| | CM-Unet | 95.11 | 91.16 | 90.89 | 79.90 | 95.93 | 94.23 | 89.27 | 0.9370 |
| | MFNet | 94.99 | 92.56 | 91.92 | 70.32 | 95.78 | 92.98 | 87.45 | 0.9346 |
| | UASFNet (Ours) | 96.50 | 95.61 | 95.12 | 78.84 | 97.29 | 95.41 | 91.52 | 0.9580 |
| Potsdam Dataset | ResNet | 95.92 | 94.24 | 94.48 | 60.41 | 95.34 | 91.6 | 86.26 | 0.9573 |
| | UNetformer | 96.46 | 95.19 | 94.39 | 61.04 | 95.51 | 91.71 | 86.77 | 0.9602 |
| | CMTFNet | 96.42 | 95.07 | 94.56 | 76.14 | 95.65 | 94.63 | 90.55 | 0.9623 |
| | CM-Unet | 89.76 | 94.74 | 92.8 | 68.84 | 96.11 | 92.43 | 86.54 | 0.9381 |
| | MFNet | 96.32 | 95.68 | 94.91 | 80.94 | 96.91 | 95.58 | 91.96 | 0.9665 |
| | UASFNet (Ours) | 96.77 | 97.64 | 94.99 | 83.84 | 97.43 | 96.37 | 93.31 | 0.9783 |
| Guiyang Dataset | ResNet | 96.35 | 96.2 | 94.46 | 66.32 | 96.27 | 93.28 | 88.33 | 0.9468 |
| | UNetformer | 96.32 | 96.22 | 94.56 | 67.86 | 96.32 | 93.56 | 88.74 | 0.9476 |
| | CMTFNet | 95.69 | 96.07 | 92.71 | 64.94 | 95.73 | 92.69 | 87.35 | 0.9392 |
| | CM-Unet | 95.65 | 96.09 | 93.76 | 65.71 | 96.00 | 92.97 | 87.80 | 0.9429 |
| | MFNet | 95.07 | 96.24 | 93.79 | 66.05 | 95.91 | 92.98 | 87.79 | 0.9419 |
| | UASFNet (Ours) | 96.58 | 96.44 | 94.62 | 67.95 | 96.45 | 93.65 | 88.90 | 0.9493 |

Fig 5 presents the land cover classification results of the Langfang Economic Development Zone for the years 2017 and 2023. In the 2017 classification results, large areas of greenbelt are observed, while built-up areas are relatively scattered and primarily concentrated in the southeastern region. By 2023, built-up land had expanded substantially, with building density increasing notably in the western, northern, and eastern parts of the city. The spatial pattern of urban development exhibited a "center-to-outskirts" expansion trend, indicating a clear acceleration of the urbanization process. Greenbelts became increasingly fragmented, particularly in the northwestern and south-central zones, where large portions of greenbelt were replaced by buildings. The 2023 classification results also reveal a clearer and more continuous road network, reflecting significant improvements in urban infrastructure connectivity.

Fig 6 illustrates the changes in the proportions of roads, buildings, greenbelts, and impervious surfaces across 34 subregions of the Langfang Economic Development Zone between 2017 and 2023. Impervious surface and greenbelt changes were derived by aggregating class-wise pixel areas from the multi-temporal classification maps. Overall, the study area experienced substantial land surface transformation, characterized by intensified imperviousness, a reduction in greenbelt, and a general expansion of built-up areas.

In terms of road coverage, most regions showed significant growth, with an average increase of approximately 37%. The most prominent increases were observed in regions 4 (+163.5%), 20 (+160.3%), and 27 (+109.6%). Building coverage exhibited a distinct polarization pattern: core development zones (e.g., regions 12, 8, 6, and 16) experienced substantial growth in built-up area, with the largest increase reaching +864.6% in region 16. In contrast, several old urban or functionally adjusted zones (e.g., regions 24, 25, and 23) showed dramatic declines, with maximum decreases exceeding −90%. Greenbelt coverage declined across all regions, with an average reduction of approximately 40%. The most pronounced decreases occurred in regions 16 (−75.9%) and 21 (−61.5%), indicating that urban expansion has exerted significant pressure on ecological spaces.

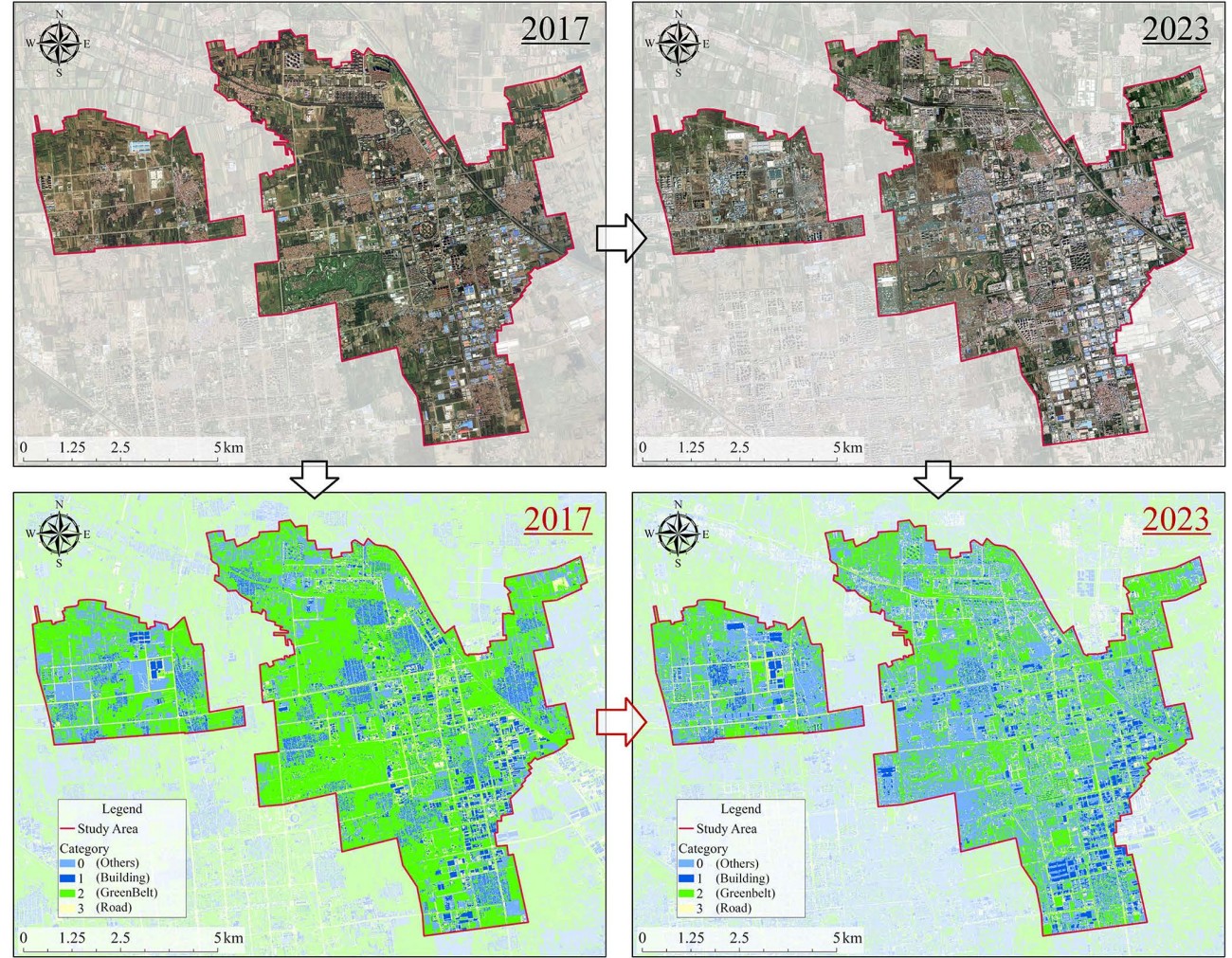

**Fig 5. Multi-temporal urban land cover classification results for the Langfang Economic Development Zone.** The background remote sensing imagery was obtained from the Copernicus Data Space Ecosystem (Copernicus Sentinel data) and processed by the authors. The imagery represents original satellite observations rather than an online basemap. All classification maps, boundaries, annotations, and map elements were generated by the authors.

Consequently, the impervious surface ratio increased across all subregions. In 2017, imperviousness ranged from 8.5% to 47%, with an average of 25.37%; by 2023, it had risen to 13%–58%, averaging 42.23%, representing an overall increase of approximately 16.86 percentage points. Spatially, the most pronounced increases in imperviousness were concentrated in the western and northern portions of the study area, reflecting a clear trend of urban development expanding outward from the city center.

### 3.3. Spatial distribution of urban land cover change evaluation indicators

Based on the multi-temporal classification results, this subsection analyzes the spatial distribution of urban land cover change evaluation indicators. The analysis focuses on the spatial patterns and heterogeneity of building, road, and greenbelt changes at the pixel and local scales.

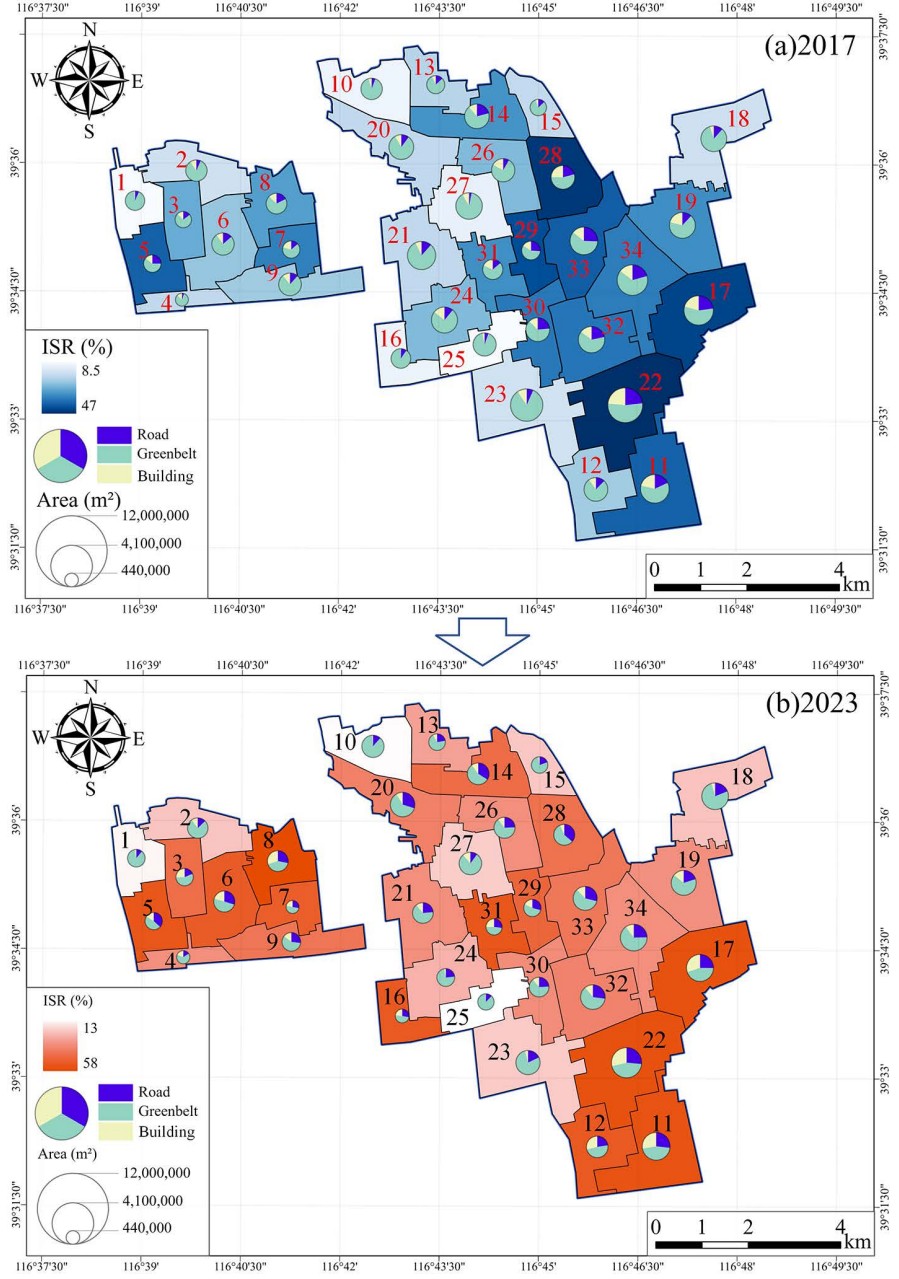

**Fig 6. Comparative charts of multiple indicators across the 34 subregions of the Langfang Economic Development Zone (2017–2023). All map elements were generated by the authors.**

The spatial distribution of urban land cover change evaluation indicators shown in Fig 7 reveals pronounced spatial differentiation characteristics of the Langfang Economic Development Zone during the urbanization process from 2017 to 2023. Overall, changes in buildings, greenbelts, and roads all exhibit distinct spatial clustering and gradient distribution patterns. Building changes are mainly concentrated in the central–southern, southeastern, and western subregions, forming a "high-in-center, low-at-periphery" spatial pattern that reflects the outward expansion of urban construction activities

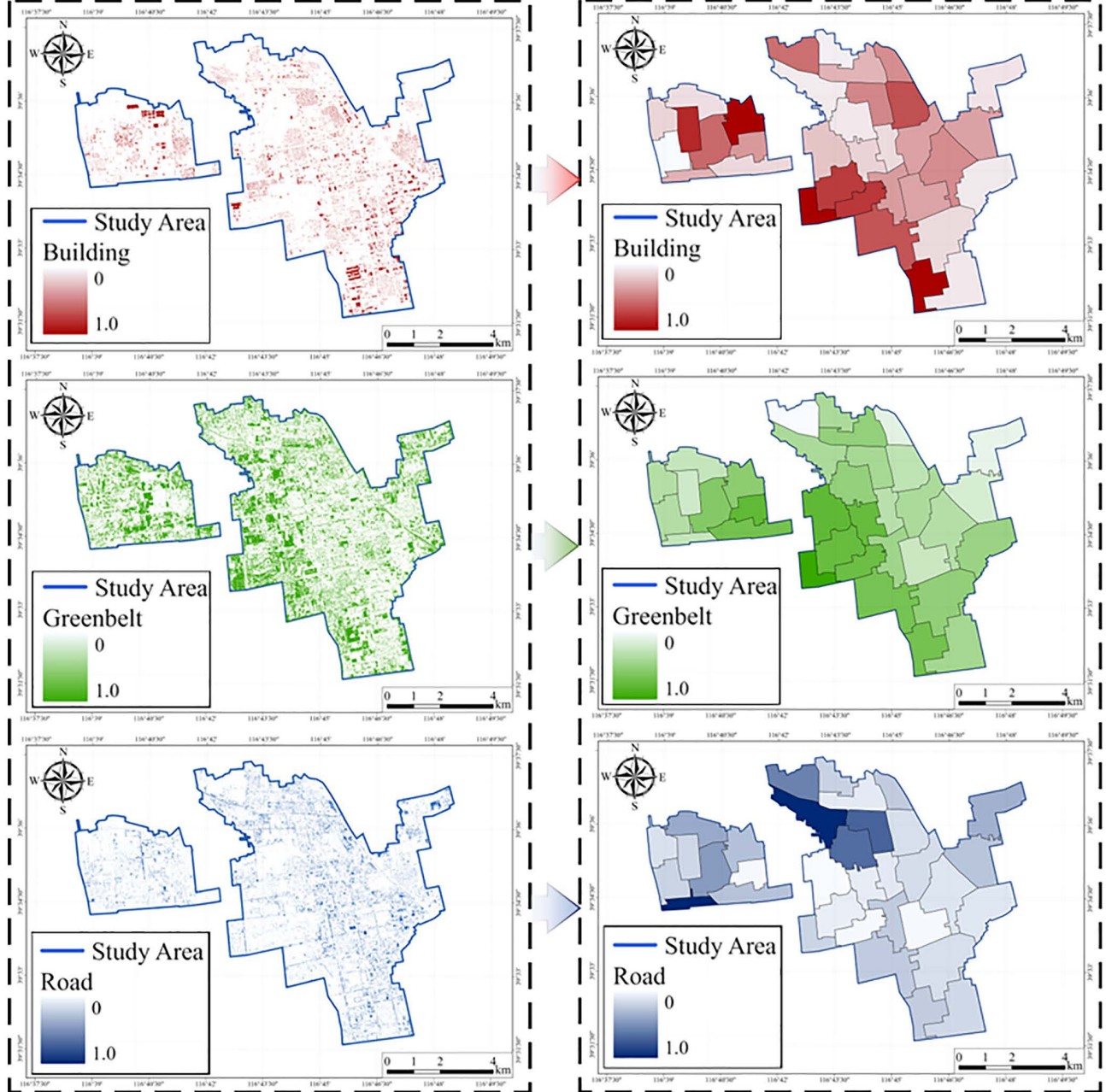

**Fig 7. Spatial distribution of individual urban land cover disturbance indicators. All thematic maps were produced by the authors.**

from the core areas. Greenbelt changes show a high degree of spatial coupling with building changes. High-value zones are primarily distributed within the core development areas and southeastern belt regions, indicating that the greenbelt system has been strongly disturbed and fragmented by urban expansion, while the ecological functions in peripheral areas remain relatively stable. The spatial pattern of road changes reflects the outward expansion of the urban transportation network. High-change areas are mainly located in the northwestern, southern, and southeastern corridors, consistent with the direction of new urban development, forming a "multi-core belt-shaped expansion" spatial configuration. Overall,

the three indicators demonstrate a synergistic spatial evolution pattern, where urban construction, road expansion, and greenbelt transformation interact to jointly shape an urban surface evolution pattern characterized by intensive development in the core areas and belt-like expansion toward the periphery. These spatially heterogeneous distribution patterns observed in Fig 7 indicate that urban land cover change is not spatially uniform but tends to concentrate in areas with intensive development activities. The co-occurrence and spatial coupling of building, road, and greenbelt disturbances suggest that urban expansion processes exhibit structured spatial organization rather than random dispersion. From an observational perspective, such spatial patterns provide useful spatial references for identifying development hotspots and relatively stable zones, which may inform future urban planning and land management decisions in terms of spatial prioritization and differentiated control.

### 3.4. Analysis of urban land cover change evaluation results

This subsection provides a comprehensive analysis of urban land cover change evaluation results by integrating multiple indicators into an overall disturbance assessment. The analysis is conducted at both pixel and regional levels to examine how localized changes aggregate into broader urban transformation patterns.

To determine the weights of urban Land Cover changes indicators, the AHP was employed to establish the relative importance among indicators. The judgment matrix in Table 3 was constructed using expert scoring and subjected to a consistency test, yielding a consistency index (CI) of 0.0193, a random consistency index (RI) of 0.52, and a consistency ratio (CR) of 0.037, which is less than 0.1. This result indicates that the constructed judgment matrix meets the consistency requirements. Consequently, the importance ranking of the indicators is as follows: building disturbance (0.637), road disturbance (0.258), and greenbelt disturbance (0.105). The pairwise comparison values were obtained through independent expert scoring following the standard AHP nine-point scale. The final judgment matrix represents the aggregated expert consensus and was verified using the consistency ratio (CR), ensuring logical consistency of the weighting scheme.

The overall disturbance distribution of the study area, calculated using Equation (10), is illustrated in Fig 8. The study area exhibits pronounced spatial heterogeneity and a distinct hierarchical pattern. High-disturbance zones are sporadically clustered in the southern, western, and southeastern construction hotspots, indicating a significant intensification of human activities at the local scale. Medium-disturbance zones are primarily located in transitional belts between the urban core and peripheral areas, corresponding to regions of active functional adjustment and spatial expansion. In contrast, low-disturbance zones are widely distributed across the central-northern and peripheral regions of the city. Overall, the disturbance intensity displays a spatial gradient from peripheral stability to core intensification, reflecting a typical urbanization development pattern. At the pixel scale, high-disturbance pixels form discrete patches within densely built-up areas, medium-disturbance pixels are distributed mainly along urban boundaries, and low-disturbance pixels constitute a continuous and stable background—collectively revealing a decreasing gradient from the urban core to the periphery. At the regional scale, the disturbance levels exhibit spatial clustering patterns: low-disturbance zones are concentrated in ecologically favorable northern and northwestern areas, while high-disturbance zones align with urban expansion corridors, forming several disturbance cores that highlight the directional nature of urbanization.

**Table 3. Expert-based AHP judgment matrix for urban land-cover disturbance indicators.**

| Evaluation index | Building disturbance | Road disturbance | Greenbelt disturbance |
|---|---|---|---|
| Building disturbance | 1 | 5 | 3 |
| Road disturbance | 1/5 | 1 | 1/3 |
| Greenbelt disturbance | 1/3 | 3 | 1 |

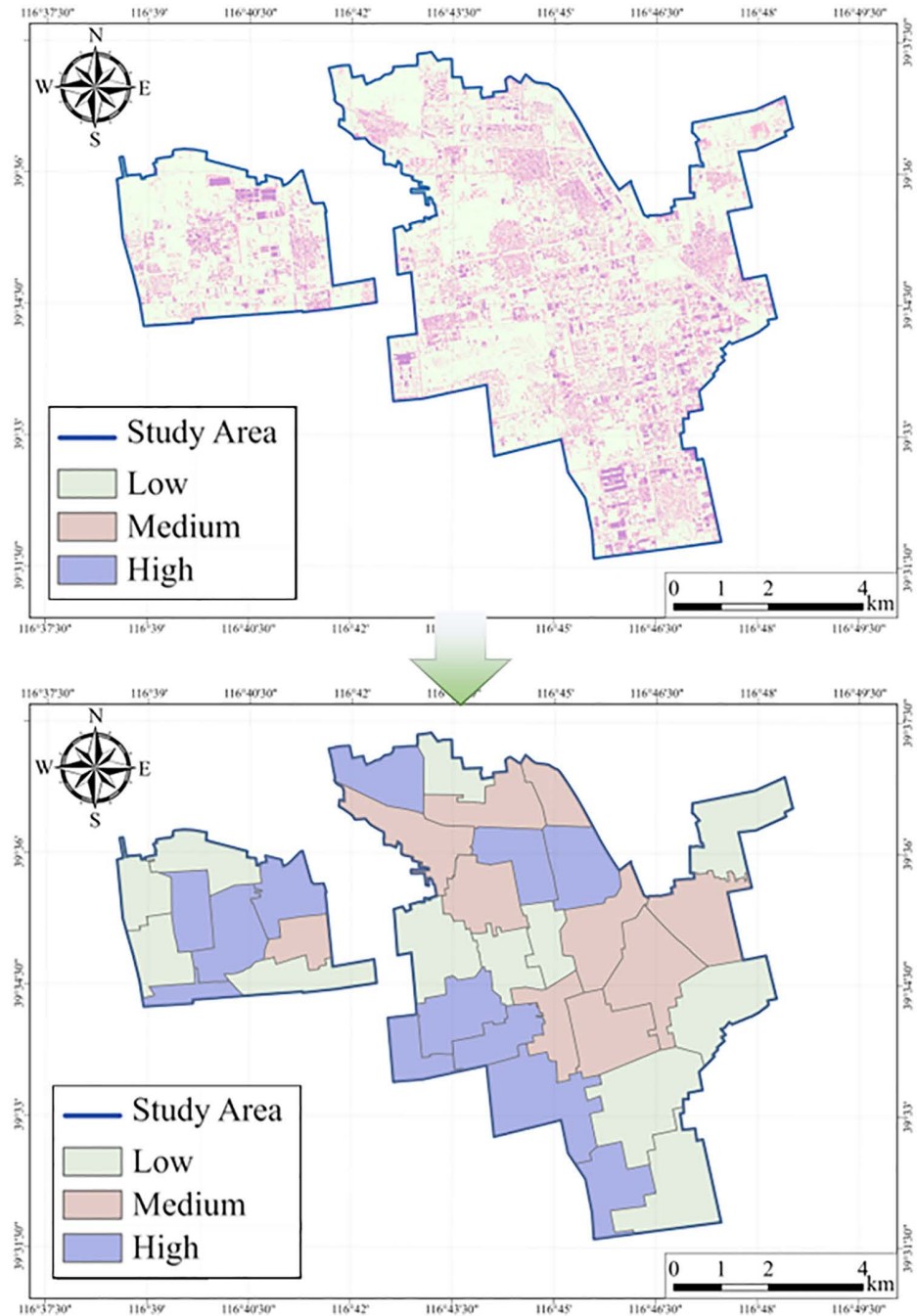

**Fig 8. Spatial distribution of comprehensive urban land cover change intensity. All visual elements were generated by the authors.**

Table 4 summarizes the degree of land cover change within the study area at both the pixel level and the regional scale. Low-disturbance pixels cover an area of 4,800.75 ha, accounting for 70.15% of the total area, while medium- and high-disturbance pixels account for 22.78% and 7.07%, respectively. This indicates that, although the overall degree of change is relatively low, it is spatially concentrated in specific localities. At the regional scale, high-disturbance zones

**Table 4. Land Cover Change Intensity at the Pixel and Regional Levels within the Study Area.**

| Disturbance level | | Low | Medium | High |
|---|---|---|---|---|
| Pixel layer | Area(ha) | 4800.75 | 1559.03 | 483.50 |
| | Scale(%) | 70.15 | 22.78 | 7.07 |
| Regional Layer | Quantity | 11 | 10 | 13 |
| | Scale(%) | 32.35 | 29.41 | 38.24 |

constitute the largest proportion (38.24%) of all regions, suggesting that although high-disturbance pixels are spatially limited, they produce a pronounced spatial aggregation effect at the regional level.

## 4. Discussion

### 4.1. Advantage analysis

This study develops a data-driven fine-scale urban land cover change assessment framework based on UASFNet, aiming to improve both recognition accuracy and interpretability in complex urban environments. Overall, the proposed framework demonstrates clear advantages in balancing segmentation performance, computational efficiency, and spatial coherence, which are critical for large-area and multi-temporal urban monitoring tasks.

From the perspective of model efficiency and performance balance, UASFNet demonstrates clear advantages over existing deep learning–based segmentation approaches. As summarized in Table 5, the proposed model achieves superior segmentation accuracy without relying on excessively large parameter sizes or computational costs. This balanced performance can be attributed to the Urban Adaptive Shared-feature Attention (UASF) module, which promotes effective semantic information sharing across multiple land cover categories while suppressing redundant feature propagation. As a result, UASFNet is able to extract more compact and discriminative representations, improving recognition performance under comparable computational conditions. All models were trained and evaluated on the same datasets and data partitions to ensure a controlled and fair comparison across different urban scenarios. The moderate increase in computational cost is justified by the consistent and non-trivial accuracy gains achieved across all datasets, which are critical for reliable large-area and multi-temporal urban change analysis.

The performance advantages of UASFNet can be attributed to the Urban Adaptive Shared-feature Attention (UASF) mechanism. This mechanism is designed to emphasize semantic and structural patterns that are consistently informative across different urban contexts, such as building boundaries, road connectivity, and greenbelt textures. Because the attention weights are learned in a data-driven manner rather than being manually specified, the mechanism can adapt to variations in urban morphology, density, and spatial configuration across different study areas. Nevertheless, its adaptability may be constrained under conditions involving pronounced domain shifts, such as cities with markedly different architectural styles, severe shadow effects, or highly heterogeneous surface materials. In such cases, incorporating additional

**Table 5. Model efficiency and performance comparative analysis.**

| Method | FLOPs (G) | Param.(M) | mIoU (%) |
|---|---|---|---|
| ResNet | 23.5 | 47.43 | 84.41 |
| UNetformer | 30.17 | 61.59 | 87.34 |
| CMTFNet | 76.41 | 96.14 | 90.30 |
| CM-Unet | 33.26 | 64.02 | 89.27 |
| MFNet | 54.39 | 67.72 | 87.45 |
| UASFNet (Ours) | 30.925 | 79.88 | 91.52 |

domain-specific supervision or multi-source auxiliary information may further enhance robustness, which is considered a direction for future work.

Beyond efficiency considerations, the spatial behavior of the proposed framework further highlights its practical advantages. The large-area recognition results from the Langfang dataset (Fig 9) illustrate that UASFNet maintains stable and coherent classification performance across extensive urban regions. Typical land cover transitions between 2017 and 2023, including urban expansion, road network evolution, and greenbelt degradation, are consistently captured. In particular, the clear delineation of impervious surfaces and linear structures indicates that the model is effective not only at the pixel level but also in preserving meaningful spatial patterns, which is essential for reliable urban change analysis.

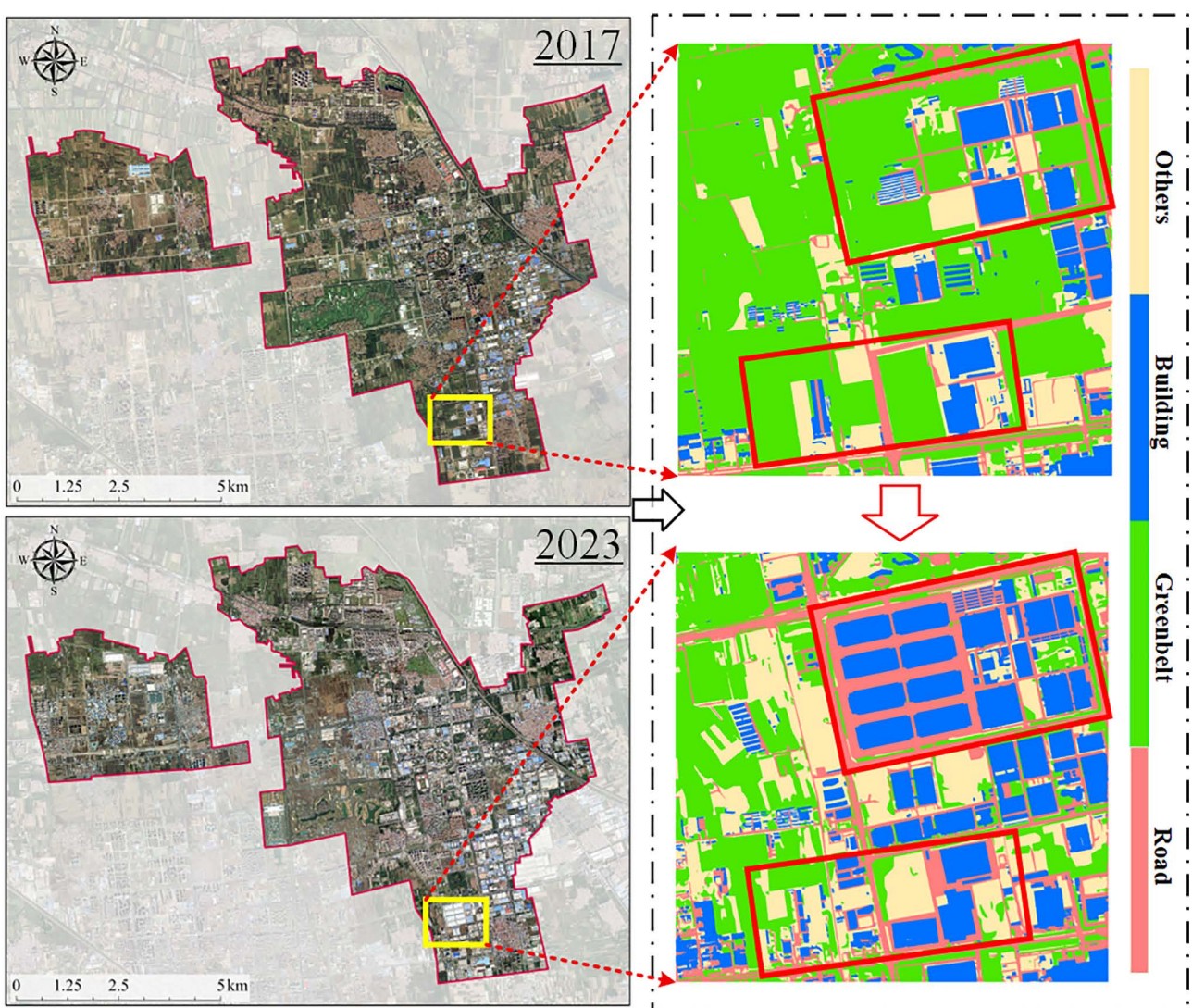

**Fig 9. Large range of recognition effect comparison diagram in Langfang. Background imagery was obtained from the Copernicus Data Space Ecosystem(Copernicus Sentinel data) and processed by the authors for visualization and comparison.** The imagery corresponds to original satellite data and is not derived from any third-party online basemap or proprietary map service. All classification results, overlays, and map elements were generated by the authors.

An additional strength of the proposed framework lies in the integration of fine-grained land cover classification with an urban land cover change evaluation scheme. By coupling high-precision pixel-level outputs with an AHP-based disturbance assessment that incorporates weighted indicators of buildings, roads, and greenbelts, the framework establishes a systematic link between classification performance and change interpretation. In this context, Table 5 and Fig 9 serve as supporting evidence demonstrating how improvements in model efficiency and spatial recognition stability contribute to more reliable inputs for quantitative urban land cover change assessment.

Overall, the advantages of the proposed framework extend beyond incremental accuracy gains. The combination of balanced computational efficiency, spatially coherent recognition, and assessment-oriented design suggests its potential applicability in scenarios such as smart city platforms, automated urban monitoring, and evidence-based spatial planning.

### 4.2. Implication and further application

The proposed framework provides meaningful implications for urban land cover change analysis and spatial governance. By enabling fine-grained land cover classification and multi-scale change assessment, it facilitates a detailed characterization of urban expansion dynamics, infrastructure development, and greenbelt transformation. Such spatially explicit information supports a more refined evaluation of land use intensity and ecological pressure during rapid urbanization processes. The integration of weighted disturbance indicators further enhances the interpretability of urban land cover change assessment. By synthesizing building, road, and greenbelt disturbances into a unified evaluation framework, the proposed approach enables consistent comparison across regions and time periods. This integrated perspective is particularly suitable for complex urban environments, where multiple land cover transitions often occur simultaneously and interactively rather than in isolation. Nevertheless, the practical application of the proposed framework may be influenced by limitations in the underlying data, such as inconsistencies in image acquisition time and variations in spatial resolution across different datasets, which could introduce uncertainty in long-term change interpretation. Addressing these data-related constraints through improved data consistency and preprocessing remains an important direction for future research. Overall, the proposed framework demonstrates strong potential for long-term urban land cover monitoring and automated change assessment. Its data-driven and modular design allows flexible adaptation to different urban contexts and data sources, providing a scalable methodological foundation for future studies on urban dynamics and spatial governance.

### 5. Conclusion

This study developed a data-driven fine-scale assessment framework for urban land cover change based on UASFNet, integrating deep learning–based multi-object recognition with Analytic Hierarchy Process (AHP) weighting to systematically characterize the spatiotemporal evolution of urban surface systems. Supported by multi-source high-resolution remote sensing data, the proposed framework combines high-precision classification with quantitative disturbance evaluation, providing a novel technical pathway for urban spatial governance and ecological optimization. The main conclusions are as follows:

(1) The proposed UASFNet model introduces an Urban Adaptive Shared-feature Attention module that enables collaborative modeling of semantic and structural features, maintaining boundary continuity and spatial consistency in the recognition of buildings, roads, and greenbelts.

(2) Across three representative datasets, UASFNet achieved mean mIoU values of 91.52%, 93.31%, and 88.90%, respectively—significantly outperforming existing representative models in the recognition accuracy of buildings, greenbelts, and roads—demonstrating strong cross-domain generalization capability.

(3) Spatial analysis indicates that during 2017–2023, high-disturbance areas in the Langfang Economic Development Zone were primarily concentrated in the southern, western, and southeastern regions, forming an urbanization gradient pattern characterized by intensive core development and peripheral belt-like expansion.

(4) The framework achieves high recognition accuracy with low computational complexity while maintaining strong interpretability and scalability, making it suitable for applications in urban impervious surface monitoring, land-use planning evaluation, and ecological resilience analysis.

## Supporting information

**S1 Table. Urban Adaptive Shared-feature Attention (UASF) Module.**
(DOCX)

## Author contributions

**Conceptualization:** Qiang Liu.

**Data curation:** Qiang Liu, Jiachen Guo, Zhixiang Da.

**Methodology:** Jiachen Guo.

**Project administration:** Qiang Liu, Feng Ling.

**Resources:** Feng Ling.

**Software:** Qiang Liu, Jiachen Guo, Chuanxing Zheng.

**Supervision:** Qiang Liu, Jiachen Guo, Chuanxing Zheng.

**Validation:** Chuanxing Zheng.

**Visualization:** Chuanxing Zheng.

**Writing – original draft:** Qiang Liu, Jiachen Guo, Chuanxing Zheng, Wenlong Song, Fengjiao Zhao, Jijian Lian.

**Writing – review & editing:** Qiang Liu, Chuanxing Zheng, Feng Ling, Zhixiang Da.

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
