## [Decision Letter · Decision Letter 0]

11 Dec 2025

Dear Dr. Zheng,

Thank you for submitting your manuscript to PLOS ONE. After careful consideration, we feel that it has merit but does not fully meet PLOS ONE’s publication criteria as it currently stands. Therefore, we invite you to submit a revised version of the manuscript that addresses the points raised during the review process.

We look forward to receiving your revised manuscript.

Kind regards,

Chong Xu

Academic Editor

PLOS One

Journal Requirements:

3. Please note that PLOS One has specific guidelines on code sharing for submissions in which author-generated code underpins the findings in the manuscript. In these cases, we expect all author-generated code to be made available without restrictions upon publication of the work. Please review our guidelines at https://journals.plos.org/plosone/s/materials-and-software-sharing#loc-sharing-code and ensure that your code is shared in a way that follows best practice and facilitates reproducibility and reuse.

5. Thank you for stating the following in the Competing Interests/Financial Disclosure section:

The authors have declared that no competing interests exist.

We note that one or more of the authors are employed by a commercial company: Tianjin Huashui Engineering Consulting Co., Ltd

6. In this instance it seems there may be acceptable restrictions in place that prevent the public sharing of your minimal data. However, in line with our goal of ensuring long-term data availability to all interested researchers, PLOS’ Data Policy states that authors cannot be the sole named individuals responsible for ensuring data access (http://journals.plos.org/plosone/s/data-availability#loc-acceptable-data-sharing-methods).

7. We note that Figures 2, 3, 4, 5, 6, 7, 8, and 9 in your submission contain map/satellite images which may be copyrighted. All PLOS content is published under the Creative Commons Attribution License (CC BY 4.0), which means that the manuscript, images, and Supporting Information files will be freely available online, and any third party is permitted to access, download, copy, distribute, and use these materials in any way, even commercially, with proper attribution. For these reasons, we cannot publish previously copyrighted maps or satellite images created using proprietary data, such as Google software (Google Maps, Street View, and Earth). For more information, see our copyright guidelines: http://journals.plos.org/plosone/s/licenses-and-copyright.

a. You may seek permission from the original copyright holder of Figures 2, 3, 4, 5, 6, 7, 8, and 9 to publish the content specifically under the CC BY 4.0 license.

Reviewers' comments:

Reviewer's Responses to Questions

**Comments to the Author**

1. Is the manuscript technically sound, and do the data support the conclusions?

Reviewer #1: Partly

Reviewer #2: Yes

Reviewer #3: Partly

Reviewer #4: Partly

2. Has the statistical analysis been performed appropriately and rigorously?

Reviewer #1: I Don't Know

Reviewer #2: I Don't Know

Reviewer #3: No

Reviewer #4: Yes

3. Have the authors made all data underlying the findings in their manuscript fully available?

Reviewer #1: No

Reviewer #2: Yes

Reviewer #3: No

Reviewer #4: Yes

4. Is the manuscript presented in an intelligible fashion and written in standard English?

Reviewer #1: Yes

Reviewer #2: Yes

Reviewer #3: Yes

Reviewer #4: Yes

Reviewer #1: 1. Expand the study to include cities with different topographies, such as mountainous cities like Guiyang.

2. On line 256, the ratio of training to validation data is not specified. Please clarify this key experimental detail.

3. For large-area identification, even with high-resolution imagery, errors are inevitable. However, the paper does not address the error characteristics of the reference data, which is critical for evaluating the reliability of the proposed method.

4. The discussion section requires revision. Since five algorithms are mentioned, it would greatly enhance the persuasiveness of your work to compare your algorithm with these existing methods using the same study areas reported in their respective papers.

Reviewer #2: Please see the attached file. Address all of my comments and gicve response to each comment. Implementing these revisions can enhance the paper's clarity, engagement, and overall contribution to the field of urban land cover change assessment.

Reviewer #3: Manuscript Title: A Fine-grained Evaluation Framework for Urban Land Cover Change Based on UASFNet Data-driven Approach

he manuscript aims to create a data-driven framework for fine-scale urban land cover change assessment using an Urban Adaptive Shared-feature Attention Network (UASFNet). The intended contribution is to support urban spatial governance and ecological optimization by revealing the dynamic evolution of surface systems during urbanization, and this framework has potential value for scientists, planners, and environmental managers who rely on accurate change detection to inform policy and restoration decisions. However, in its current form, the manuscript does not sufficiently articulate testable scientific questions, nor does it clarify how the proposed framework is evaluated relative to existing methods or how each methodological component contributes to measurable improvements. In addition, while the abstract presents what it calls “experimental results,” the manuscript contains no dedicated Results section—only a Case Study—which makes it difficult to understand how findings were derived or how performance was objectively assessed. In my opinion, the Case Study should be removed, with its material reused so that its computational details are incorporated into the Methods while its model comparisons and land-cover disturbance findings are placed into a properly structured Results section. Strengthening these elements collectively would improve clarity, reproducibility, and the interpretability and impact of the work.

Major Comments by Section

Title:

The acronym UASFNet does not convey meaning to most readers and may obscure the paper’s intent. A more reader-focused alternative would be: “A Fine-grained Evaluation Framework for Urban Land Cover Change Based on Feature Monitoring with Remotely Sensed Imagery.” This revision communicates the achievement rather than emphasizing a technical acronym unfamiliar to the journal’s broad readership.

Abstract:

The abstract outlines the general workflow but omits crucial information about the type of remote sensing imagery required. Given the diversity of sensors, platforms, pixel sizes, and spectral characteristics, this omission limits interpretability and reproducibility.

The following components from the Case Study section should be incorporated into the abstract to clarify imagery requirements: Google Earth RGB imagery, 0.6 m resolution (Langfang dataset); Ultra–high-resolution TOP imagery, 5 cm GSD, multiple spectral combinations (Potsdam dataset); Standardized NAIP imagery for additional regional evaluation. Adding this information makes clear that the approach requires sub-meter to centimeter-scale RGB (and optionally IR) imagery characteristic of urban remote sensing segmentation tasks.

Introduction:

The final paragraph presents the study’s intended academic contributions, but these are not framed as testable scientific questions. For example, the introduction could explicitly pose questions such as:

Does UASFNet significantly outperform established semantic segmentation methods (e.g., ResNet, UNetformer, MFNet) in boundary preservation and inter-class separability for urban land cover types?

Does integrating AHP-derived indicator weights meaningfully improve the spatial coherence or interpretability of disturbance-level assessments?

Can combined pixel-level and region-level analyses provide a measurably more accurate or actionable representation of urban land cover change than existing pixel-only approaches?

Explicit questions would help readers understand what hypotheses the study is designed to test and how success is measured.

Methods:

sub-heading UASFNet Mathematical Formulation- The core equations for the UASF module—semantic–structural consistency kernel, gradient-domain formulation, and global gating operator—are insufficiently explained for a general audience. The manuscript introduces functions (ϕ(X), ϕ(∇X), Θ(X), A(X)) without an illustrative example.

The authors should include:

A worked example: Starting from a raw image tile, show how features, gradients, embeddings, and fused representations are extracted and transformed through each stage.

A schematic diagram showing how semantic and structural pathways merge and how gating weights are computed.

Code availability: Since PLOS ONE emphasizes reproducibility, the authors should provide code or pseudocode to clarify the transformations embodied in Equations (2) and (3).

Without such explanation, the method is not reproducible or interpretable by most readers.

sub-heading Evaluation Framework - The manuscript states that “multiple metrics” are used—mIoU, accuracy, F1, Kappa, etc.—but does not clearly tie these metrics to specific stated objectives (boundary preservation, inter-class separability, semantic consistency). The connection between objectives and metrics must be explicit. Similarly, the land cover change evaluation framework (using AHP to weight building, road, and greenbelt disturbance indices) would benefit from: A clear workflow diagram; A rationale for indicator selection; An explanation of how errors in land cover classification propagate into disturbance-level scoring.

Case Study Section

1. Section 3.2: “Experimental Detail”. This subsection belongs more naturally in Methods under a heading such as: “Computational Environment and Data Augmentation Strategy.”

The description of the GPU, Python environment, and augmentation techniques is overly brief and should be expanded with parameter values and more details for helping others evaluate, test, replicate, improve your work. There are standard components of deep learning methodology and are necessary for reproducibility.

2. Model Performance (currently part of Case Study)

This section actually constitutes Results, not a case study. The models used for comparison—ResNet, UNetformer, CMTFNet, CM-Unet, MFNet—should be introduced earlier in the Methods section under: “Benchmark Models and Comparative Evaluation Design.” Currently, the paper reads as though the evaluation details appear only after results, which makes the structure confusing.

3. Case Study Results: Land cover change interpretation

The manuscript presents tables of pixel-level and region-level disturbance statistics and interprets the spatial implications, but this should be more clearly labeled as Results, not embedded inside the case study narrative. The interpretation itself is appropriate and meaningful, showing both fine-scale and aggregated change patterns.

Objectivity of Performance Evaluation

Strengths: The manuscript compares UASFNet to five baseline models across three datasets (Langfang, Potsdam, NAIP). Metrics (mIoU, F1, Kappa) are standard for segmentation research. Improvements reported are consistent (≈1–3% advantage across categories).

Weaknesses: The evaluation is not explicitly tied to the stated objectives (e.g., boundary preservation, structural consistency). No statistical tests (e.g., paired t-tests on pixel accuracy, variance analyses across tiles) are reported to support claims of significance. The study does not examine failure cases, computational complexity trade-offs, or robustness to noise, which would strengthen claims of generalizability. The evaluation does not address whether the AHP-based disturbance metrics produce more accurate or actionable land cover change assessments than alternative weighting methods.

Conclusion on Objectivity: The manuscript does provide a comparative performance assessment, but it is only partially objective and not fully aligned with the evaluation goals described in the Methods section. Strengthening the study would require tighter linkage between model objectives and metrics, inclusion of statistical validation, and clearer articulation of how the proposed framework improves land cover change interpretation beyond classification accuracy alone.

Reviewer #4: Overall Comments

This paper presents a data mining-based method for identifying urban area changes from remote sensing images. The manuscript is generally well-written. The proposed method has been implemented on several areas and its performance has been compared with several common methods. However, revisions are necessary before publication.

Major Comments

1. Introduction/Literature Review: While a general categorization of research in the field is provided, a more focused review of studies directly related to the presented methodology is needed. This would establish a clearer baseline and context, allowing readers to better assess the novelty and contribution of the proposed method relative to existing work.

2. Research Objectives: The objectives stated at the end of the introduction should more explicitly highlight and emphasize the novel aspects of the proposed approach compared to previous studies.

3. Methodology Description (Clarity & Flow):

o Lines 111-113 / Overall Procedure: The description of the proposed method is somewhat vague. The specific steps of the pipeline should be clearly delineated. For each stage, it should be explicitly stated what processing is applied to the input, what the output is, and why this output is fed into the next stage. This lack of clarity is also reflected in the provided flowchart, making it difficult for the reader to follow the logical flow of the method.

o Justification of Weighting: The rationale and importance of using a weighting scheme (Lines 111-113, Line 180) are not sufficiently explained. A clear justification is required: Why is weighting crucial for identifying the target changes in this specific context? What problem does it solve that non-weighted approaches do not?

4. Methodology Subsections & Citations: The subsections under the methodology present various formulas and relationships without any citations to prior work. It appears that all elements were developed by the authors, which is unusual for a standard data mining/remote sensing workflow. Relevant sources for established techniques should be cited.

5. Subsection Structure (Lines 234-236): The inclusion of several very brief subsections with minimal content is not effective. Each subsection should contain substantive material related to its title.

Specific Comments on Figures and Tables

• Figure 1: The caption should be more detailed and specific, describing what the figure illustrates. This applies to all figure and table captions throughout the manuscript.

• Figure 3: The defined legend for this figure appears to be incorrect. Please verify and correct it.

• Table 2: It is unclear why all values for the proposed method are highlighted with a different color. Typically, only the best-performing values (e.g., highest accuracy) should be bolded or highlighted for clear comparison.

• Table 3: The methodology for generating the rankings is not described. How were these rankings obtained? Was a single expert used, or multiple experts? If multiple, how was consensus reached? This information is essential for assessing the validity of the comparison.

Recommendation

The paper addresses an interesting topic but requires significant revisions, primarily to improve the clarity, justification, and context of the proposed methodology, and to ensure rigorous presentation of results and comparisons. Addressing these points will strengthen the manuscript considerably.

**Do you want your identity to be public for this peer review?** For information about this choice, including consent withdrawal, please see our Privacy Policy

Reviewer #1: No

Reviewer #2: **Yes:** Muhammad Nasar AHMAD

Reviewer #3: No

Reviewer #4: **Yes:** Milad Janalipour

---

## [Author Response · Author response to Decision Letter 1]

12 Jan 2026

Thanks for all comments and suggestions of the reviewers. They are all significant for our research work and paper writing. We have a detailed revision of this paper and changes are marked in yellow. The presentation in the revised paper has been improved. Now we will response to the comments as follows.

Response to reviewer #1:

(1)Q: Expand the study to include cities with different topographies, such as mountainous cities like Guiyang.

A: We sincerely appreciate this valuable suggestion. We agree that including cities with diverse topographies is crucial for validating the robustness and generalization ability of our method. Following your advice, we have expanded our study to include the Guiyang Dataset, which represents a typical mountainous urban environment. This addition ensures our study covers a broader range of geographical scenarios.

The revisions have been made in Section 2.2, 3.1 (Lines 147-150, 348-353) of the revised manuscript. The added description is as follows:.

“(c) Guiyang Dataset: This dataset covers the urban core area of Guiyang City, China, representing a typical mountainous urban environment. It was constructed following the same data preparation, annotation protocol, and data partition strategy as the Langfang dataset, ensuring consistency and comparability across datasets.

On the Guiyang dataset, UASFNet achieved the best overall performance, with an mIoU of 88.90%, an F1-score of 93.65%, and a Kappa value as high as 0.9493. Among the four land-cover categories, the IoU values for buildings, roads, and greenbelts reached 96.58%, 96.44%, and 94.62%, respectively.”

(2)Q: On line 256, the ratio of training to validation data is not specified. Please clarify this key experimental detail.

A: We thank the reviewer for pointing out this oversight. We apologize for not specifying the ratio of training to validation data in the original manuscript. To address this, we have explicitly clarified the data partition strategy and the specific quantity of data for both the Langfang and Potsdam datasets in the revised version.

Specifically, we specified a ratio of 70% / 10% / 20% (training/validation/test) for the Langfang dataset and provided the exact number of image patches for the Potsdam dataset. The revisions can be found in Section 2.2.2 (Lines 136-146) of the revised manuscript:

“(a) Langfang Dataset: This dataset was constructed using RGB three-band remote sensing images of the Langfang Economic Development Zone with a spatial resolution of 0.6 m. Four land-use classes were annotated using GIS software, and the dataset was partitioned into training, validation, and test sets using a spatial block segmentation method at a ratio of 70% / 10% / 20%, respectively.

(b) Potsdam Dataset: This dataset is derived from ultra–high-resolution TOP imagery with a ground sampling distance (GSD) of 5 cm. The Potsdam region is known for its complex building layouts and dense urban structures. The dataset covers an area of 3.42 km² and includes pixel-level annotations for six semantic categories. It has become a standard benchmark for semantic segmentation research. In this study, an improved four-class semantic labeling scheme—including buildings, greenbelts, roads, and others—was adopted to better suit urban analysis tasks, resulting in 2299 training, 605 validation, and 1694 test image patches of size 1024 × 1024.”

(3)Q: For large-area identification, even with high-resolution imagery, errors are inevitable. However, the paper does not address the error characteristics of the reference data, which is critical for evaluating the reliability of the proposed method.

A: This is a highly insightful and critical comment. We fully agree with the reviewer that addressing the error characteristics of the reference data is essential for properly evaluating the reliability of the proposed method. To address this concern, we have added a discussion acknowledging that, as with most large-area urban mapping studies, reference labels are subject to unavoidable uncertainties, particularly in transitional zones, complex building boundaries, and shadow-affected areas. Furthermore, we clarified that since all comparative models were trained and evaluated using the same reference data and partitioning strategy, these uncertainties affect all methods consistently and do not compromise the validity of the relative performance comparison.

The revisions can be found in Section 2.4 (Lines 230-234) of the revised manuscript:

“It should be noted that, as in most large-area urban land cover mapping studies, reference labels are subject to unavoidable uncertainties, particularly in transitional zones, complex building boundaries, narrow road segments, and shadow-affected areas. Since all models were trained and evaluated using the same reference data and partitioning strategy, such uncertainties affect all methods consistently and do not compromise the validity of the relative performance comparison.”

(4)Q: The discussion section requires revision. Since five algorithms are mentioned, it would greatly enhance the persuasiveness of your work to compare your algorithm with these existing methods using the same study areas reported in their respective papers.

A: We greatly appreciate the reviewer’s suggestion regarding the comparative analysis. We understand that evaluating algorithms on their original study areas can provide context. However, direct comparisons across different original datasets can introduce significant bias due to variations in image sources, resolutions, annotation standards, and partition strategies.

To ensure the most rigorous and fair comparison, our strategy was to evaluate all five algorithms under identical experimental settings (using the same datasets and data partition protocols). This approach eliminates the interference of data heterogeneity and allows for a more accurate assessment of the intrinsic performance differences between the models. We have clarified this rationale in the revised manuscript to emphasize the validity of our controlled comparison. The revisions can be found in Section 4.1 (Lines 488-489) of the revised manuscript:.

“All models were trained and evaluated on the same datasets and data partitions to ensure a controlled and fair comparison across different urban scenarios.”

Response to reviewer #2:

(1)Q: All figures should be improved in term of map elements delivery and clarity. For each figure, include clear captions contextualizing the data presented, methodologies used for analysis, and key takeaways. e.g. Fig. 3 it is difficult to identify difference between others and road class. I suggest mark road as red/grey and others class as sand color. Figure 1 is difficult to understand. Only there are two steps highlighted, flow is not symmetrical.

A: Thank the reviewers for their detailed suggestions regarding the quality of the charts, we have redrawn the relevant diagrams in the text, with a particular focus on enhancing the expression of map elements and improving the overall clarity. The specific modifications are as follows:

Regarding Figure 1: We have redesigned the structure of the flowchart, optimized the highlighting of steps, and ensured that the overall process is more symmetrical and the logic is more coherent, so that readers can understand our technical route more easily.

Regarding Figure 3: We have adopted your suggestion and adjusted the color scheme, enhancing the contrast between the categories (especially between the roads and other categories) to ensure a clear visual distinction.

The updated figures and captions can be found in Section 2 of the revised manuscript. The revised captions are as follows:.

Fig. 1 Overall framework of the proposed fine-scale urban land cover change assessment method.

Fig. 3 Example image patches and corresponding ground truth labels from the Potsdam dataset.”

(2)Q: How does the proposed UASFNet model compare to traditional methods in terms of accuracy and computational efficiency?

A: We thank the reviewer for this important question. we have provided a detailed comparative analysis in the revised manuscript (see Table 5). The results indicate that, from the perspective of model efficiency and performance balance, UASFNet demonstrates clear advantages over existing deep learning–based segmentation approaches. Specifically, our model achieves superior segmentation accuracy without relying on excessively large parameter sizes or computational costs. This balanced performance is primarily attributed to the Urban Adaptive Shared-feature Attention (UASF) module, which promotes effective semantic information sharing across multiple land cover categories while suppressing redundant feature propagation. As a result, UASFNet is able to extract more compact and discriminative representations, improving recognition performance under comparable computational conditions.

The revisions and detailed discussions can be found in Section 4.1 (Lines 481-488) and Table 5 of the revised manuscript..

“From the perspective of model efficiency and performance balance, UASFNet demonstrates clear advantages over existing deep learning–based segmentation approaches. As summarized in Table 5, the proposed model achieves superior segmentation accuracy without relying on excessively large parameter sizes or computational costs. This balanced performance can be attributed to the Urban Adaptive Shared-feature Attention (UASF) module, which promotes effective semantic information sharing across multiple land cover categories while suppressing redundant feature propagation. As a result, UASFNet is able to extract more compact and discriminative representations, improving recognition performance under comparable computational conditions.

Table 5 Model Efficiency and Performance Comparative Analysis

Method FLOPs (G) Param.(M) mIoU (%)

ResNet 23.5 47.43 84.41

UNetformer 30.17 61.59 87.34

CMTFNet 76.41 96.14 90.30

CM-Unet 33.26 64.02 89.27

MFNet 54.39 67.72 87.45

UASFNet (Ours) 30.925 79.88 91.52

(3)Q: In 2.1 Fine-Scale Assessment Framework (Lines 106-113), what specific preprocessing techniques were applied to the bi-temporal remote sensing images? Could this impact the overall accuracy of the analysis?

A: We thank the reviewer for the inquiry regarding preprocessing details. We agree that specifying these steps is essential for reproducibility and reliability.

In the revised manuscript, we have explicitly detailed the preprocessing techniques applied. Specifically, to ensure consistency across multi-source and multi-temporal imagery, we performed geometric co-registration, spatial resampling to a unified resolution, and RGB value normalization. Regarding the potential impact on accuracy, we clarified that we deliberately avoided aggressive radiometric correction to preserve original spatial patterns. Furthermore, by applying this identical preprocessing pipeline to all datasets and baseline models, we minimize bias and ensure a fair and unbiased comparison.

The revisions can be found in Section 2.2.2 (Lines 151-157) of the revised manuscript:

“For all datasets, standard preprocessing procedures were applied prior to model training and inference to ensure consistency across multi-source and multi-temporal imagery. These procedures include geometric co-registration between different acquisition periods, spatial resampling to a unified spatial resolution, and normalization of RGB values. No aggressive radiometric correction or handcrafted feature extraction was introduced, in order to preserve original spatial patterns. The same preprocessing pipeline was applied to all datasets and baseline models to ensure a fair and unbiased comparison.”

(4)Q: For 2.2 UASFNet Model (Lines 117-164), how does the shared-feature attention mechanism operate in varying urban contexts? Are there limitations to its adaptability?

A: We thank the reviewer for this insightful question regarding the core mechanism of our model. Discussing the adaptability and boundary conditions of the attention mechanism is crucial for understanding the model's reliability.

In the revised manuscript, we have elaborated on the operational principle of the UASF mechanism: it operates in a data-driven manner to learn attention weights, thereby automatically focusing on semantic and structural patterns that remain consistent across varying urban contexts (such as building boundaries, road connectivity, and greenbelt textures).

In the discussion section, relevant explanations were also provided. When encountering significant "domain shifts" (such as huge differences in architectural styles, severe shadow effects, or highly heterogeneous surface materials), the adaptability of the model may be limited.

The revisions and discussion can be found in Section 4.1 (Lines 493-502) of the revised manuscript:

“The semantic–structural consistency kernel encourages the alignment of semantic representations with geometric cues, which is beneficial for preserving object boundaries and reducing confusion between visually similar land cover classes. The gradient-domain formulation highlights local structural variations, such as edges and shape transitions, that are particularly relevant for fine-scale urban patterns. The global gating operator adaptively modulates the relative contributions of semantic and structural information across spatial locations, thereby promoting spatial coherence while limiting redundant feature propagation.

The performance advantages of UASFNet can be attributed to the Urban Adaptive Shared-feature Attention (UASF) mechanism. This mechanism is designed to emphasize semantic and structural patterns that are consistently informative across different urban contexts, such as building boundaries, road connectivity, and greenbelt textures. Because the attention weights are learned in a data-driven manner rather than being manually specified, the mechanism can adapt to variations in urban morphology, density, and spatial configuration across different study areas. Nevertheless, its adaptability may be constrained under conditions involving pronounced domain shifts, such as cities with markedly different architectural styles, severe shadow effects, or highly heterogeneous surface materials. In such cases, incorporating additional domain-specific supervision or multi-source auxiliary information may further enhance robustness, which is considered a direction for future work.”

(5)Q: (Lines 165-171) Why were the specific evaluation metrics chosen, and do you believe they comprehensively reflect the model's performance?

A: We thank the reviewer for the question regarding the selection of evaluation metrics. We fully agree that a well-justified combination of metrics is essential for a comprehensive and objective assessment of model performance.

To address this, we have elaborated on the scientific rationale behind our choices in the revised manuscript. Specifically, we employed mIoU as the primary metric for quantifying overall pixel-level accuracy. This is complemented by Precision, Recall, and F1-score to characterize class-wise discrimination, which is particularly critical for handling imbalanced or structurally complex categories common in urban scenes. Furthermore, we included the Kappa coefficient to evaluate the agreement between predicted and reference maps beyond random chance.

We believe that this suite of metrics comprehensively reflects the model's performance by capturing pixel-wise accuracy, class-wise consistency, and statistical robustness. This ensures that the resulting land-cover maps serve as reliable inputs for the subsequent urban disturbance analysis. The revisions can be found in Section 2.4 (Lines 217-224) of the revised manuscript:

“The mean Intersection over Union (mIoU) was used as the primary metric to quantify overall pixel-level classification accuracy across land-cover classes. Precision, recall, and F1-score were employed to further characterize class-wise discrimination performance, particularly for imbalanced or structurally complex categories. In addition, Kappa coefficient was included to evaluate the agreement between predicted and reference maps beyond chance, providing a robust measure of classification reliability. Together, these metrics capture not only pix

---

## [Decision Letter · Decision Letter 1]

22 Jan 2026

A fine-grained evaluation framework for urban land cover change based on feature monitoring with remotely sensed imagery

PONE-D-25-61989R1

Dear Dr. Zheng,

We’re pleased to inform you that your manuscript has been judged scientifically suitable for publication and will be formally accepted for publication once it meets all outstanding technical requirements.

Kind regards,

Chong Xu

Academic Editor

PLOS One

Additional Editor Comments (optional):

Reviewers' comments:

Reviewer's Responses to Questions

**Comments to the Author**

Reviewer #1: All comments have been addressed

Reviewer #3: All comments have been addressed

2. Is the manuscript technically sound, and do the data support the conclusions?

Reviewer #1: (No Response)

Reviewer #3: Yes

3. Has the statistical analysis been performed appropriately and rigorously?

Reviewer #1: (No Response)

Reviewer #3: Yes

4. Have the authors made all data underlying the findings in their manuscript fully available?

Reviewer #1: (No Response)

Reviewer #3: Yes

5. Is the manuscript presented in an intelligible fashion and written in standard English?

Reviewer #1: (No Response)

Reviewer #3: Yes

Reviewer #1: (No Response)

Reviewer #3: I would like to thank the authors for their careful responses to my prior comments. The revisions have strengthened the manuscript. The goal was to make the contribution easier to interpret, reproduce, and evaluate relative to existing approaches.

**Do you want your identity to be public for this peer review?** For information about this choice, including consent withdrawal, please see our Privacy Policy

Reviewer #1: No

Reviewer #3: No

---

## [Editor Report · Acceptance letter]

PONE-D-25-61989R1

PLOS One

Dear Dr. Zheng,

I'm pleased to inform you that your manuscript has been deemed suitable for publication in PLOS One. Congratulations! Your manuscript is now being handed over to our production team.

Kind regards,

on behalf of

Dr. Chong Xu

Academic Editor

PLOS One